REGISTERED REPORT PROTOCOL

# The impact of visually simulated self-motion on predicting object motion–A registered report protocol

**Björn Jörges***, **Laurence R. Harris**

Center for Vision Research, York University, Toronto, Canada

* bjoern_joerges@hotmail.de

## Abstract

To interact successfully with moving objects in our environment we need to be able to predict their behavior. Predicting the position of a moving object requires an estimate of its velocity. When flow parsing during self-motion is incomplete–that is, when some of the retinal motion created by self-motion is incorrectly attributed to object motion–object velocity estimates become biased. Further, the process of flow parsing should add noise and lead to object velocity judgements being more variable during self-motion. Biases and lowered precision in velocity estimation should then translate to biases and lowered precision in motion extrapolation. We investigate this relationship between self-motion, velocity estimation and motion extrapolation with two tasks performed in a realistic virtual reality (VR) environment: first, participants are shown a ball moving laterally which disappears after a certain time. They then indicate by button press when they think the ball would have hit a target rectangle positioned in the environment. While the ball is visible, participants sometimes experience simultaneous visual lateral self-motion in either the same or in the opposite direction of the ball. The second task is a two-interval forced choice task in which participants judge which of two motions is faster: in one interval they see the same ball they observed in the first task while in the other they see a ball cloud whose speed is controlled by a PEST staircase. While observing the single ball, they are again moved visually either in the same or opposite direction as the ball or they remain static. We expect participants to overestimate the speed of a ball that moves opposite to their simulated self-motion (speed estimation task), which should then lead them to underestimate the time it takes the ball to reach the target rectangle (prediction task). Seeing the ball during visually simulated self-motion should increase variability in both tasks. We expect to find performance in both tasks to be correlated, both in accuracy and precision.

## Introduction

We are constantly immersed in complex, dynamic environments that require us to interact with moving objects, for example when passing, setting, or hitting a volleyball, or when

---

**Data Availability Statement:** All scripts and all data obtained with regards to this project will be

made available in the project GitHub repository (https://github.com/b-jorges/Predicting-while-Moving/). Larger files such as the programs used to present stimuli and the respective Unity projects are hosted on OSF (https://osf.io/eayf7/).

**Funding:** BJ and LRH are supported by the Canadian Space Agency (CSA) (CSA: 15ILSRA1-York). LRH is supported by a Discovery Grant from the Natural Sciences and Engineering Research Council (NSERC) of Canada (NSERC: RGPIN-2020-06093). The funders did not play any role in the study design, data collection and analysis, decision to publish, or preparation of the manuscript.

**Competing interests:** The authors declare no competing interests.

deciding whether we can make a safe left turn before another car reaches the intersection. In such situations, it can often be important to predict how objects in our environment will behave over the next few moments. Predictions allow us, for example, to time our actions accurately despite neural delays [1–3] in perceiving moving objects and issuing and executing motor commands [4, 5]. Delays of between 100ms and 400ms between visual stimulation and motor response are generally assumed [6]. Without predicting or anticipating motion, we would thus always be acting on outdated positional information and be doomed to fail when attempting to intercept moving objects, particularly when they are relatively fast. Further, and perhaps more obviously, predictions are also important when moving objects are occluded during parts of their trajectory [7–9], or when the observer has to avert their eyes or turn away from the target.

When an observer is moving while attempting to interact with moving objects in their environment, further difficulties arise. Even in the simplest case, when the observer has a good view of the target and predictions are only necessary to compensate for neural delays, the visual system needs to separate retinal motion created by observer motion from retinal motion due to object motion in order to judge an object's trajectory accurately. A prominent hypothesis on how this is achieved is the Flow Parsing hypothesis [10–16]. This hypothesis states that to solve this problem humans parse optic flow information and subtract this visual stimulation attributed to self-motion from global retinal motion. The remaining retinal motion can then be scaled with an estimate of the distance between the observer and the moving object [17] to obtain the world-relative velocity of the object. While this process was originally posited as a purely visual phenomenon [18, 19], more recent studies have shed some light on the multisensory nature of this process: Dokka and her colleagues [20] found, for example, that compensation for self-motion in speed judgements was more complete when both visual and vestibular information indicated self-motion than when either visual or vestibular cues indicated that the observer was at rest. They further found precision to be lower in the presence of self-motion, which they attributed to self-motion information being noisier than optic flow information. Subtracting noisy self-motion information [21] from less noisy optic flow [20] would then make the estimate of object motion noisier in a moving observer than in a static observer. It is important to note that such a flow parsing mechanism should be active even while the participant at rest. The assumption here is that the noise added during flow parsing is proportional to the self-motion speed (in a Weber's Law-like fashion), that is, when the observer is not moving at all the added noise is minimal, whereas higher self-motion speeds lead to more noise.

Several studies have shown that flow parsing is often incomplete [9, 22–27]. A candidate explanation for this incompleteness is an inaccurate estimation of self-motion: typically, studies have used stimuli where only some cues indicated that the observer was moving (usually visual and/or vestibular cues), while efferent copy cues indicated that their bodies were at rest. If all self-motion cues are integrated after weighting them (e.g., according to their relative reliabilities [21, 28, 29]), this would then lead to an underestimation of self-motion, which is consistent with the biases found in the studies cited above. Further, while we did not find evidence for an effect of self-motion on precision in a recent study [30], we believe it is likely due to the fact that the effect was noisier than anticipated, resulting in a lack of statistical power.

Fig 1 shows a simple schematic of the processes we assume to be at play when predicting object motion during self-motion: the organism first estimates its own motion in the environment from the various sources of information available to it. Based on this self-motion estimate, the organism then makes a prediction about the retinal motion that this self-motion would be expected to create. The predicted retinal motion is then subtracted from the total observed retinal motion and any remaining motion is attributed to the object, a process we call "multisensory flow parsing" to distinguish it from the purely visual conceptualization of flow

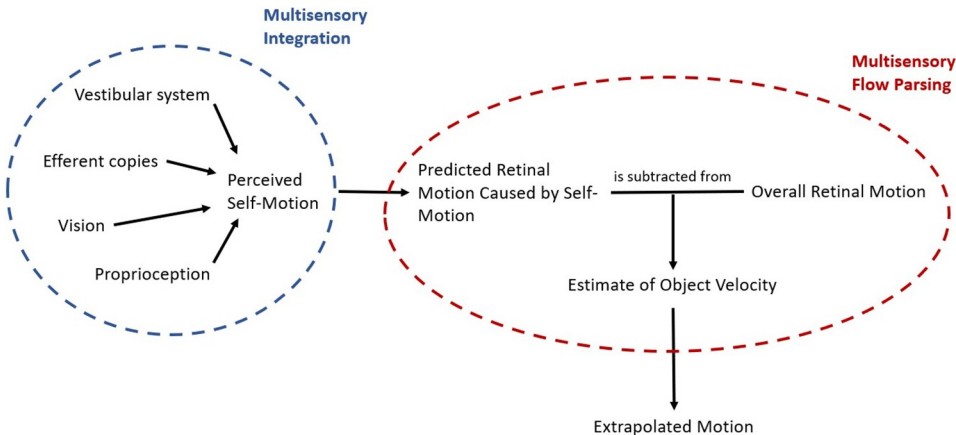

**Fig 1. Schematic of the processes at play when predicting motion during self-motion.** First, an estimate of the self-motion velocity is obtained by integrating the available cues from different sensory sources. This velocity estimate is used to predict the retinal motion that would be caused by self-motion. Finally, an estimate of the physical object velocity is obtained by subtracting the predicted retinal motion due to self-motion from the global retinal motion and the further trajectory of the object is extrapolated based on this estimate.

parsing brought forward by authors like Wertheim [15, 16] or, more recently, Dupin and Wexler [10]. This step adds noise to the velocity estimate because the self-motion estimate is noisier than the retinal input [20]. The further trajectory of the object would then be extrapolated based on this estimate of its velocity.

This schematic provides an overview over the general mechanism that might be at play when predicting future motion of an object observed during self-motion. Depending on the motion profiles of observer and object further complications may arise. For example, retinal speeds depend not only on the physical speeds and directions of the observer and object but also on the distance between them and can therefore change systematically even when observer and object move at constant physical speed without changing direction. To obtain a veridical representation of the physical velocity of the object, the observer thus has to perform additional computations, including estimating their distance to the object, the direction of the object in relation to their own direction of motion, and the necessary transformations to obtain the physical object speed from these values [17].

Some studies suggest that biases incurred while estimating motion, e.g., due to the Aubert-Fleischl effect which lowers the perceived speed of a target when an observer tracks it with their gaze [31], or due to low contrast in the stimulus [32], might transfer to biases in motion extrapolation based on these speed estimates. It seems straight-forward that any biases and precision differences observed in the perception of speed would correlate perfectly with errors and precision differences in time-to-contact judgements. However, there are two complications: first, participants might integrate biased and less precise speed information obtained during self-motion with prior information they have formed in response to previous exposure to the stimulus. They might thus extrapolate motion biased on a combination of prior information and (biased and more variable) online information. Further, it has been reported that under certain circumstances perceptual judgements and action-related tasks can be based on separate cues (see, e.g., [33–35]). While it remains an appealing hypothesis, it should thus not be assumed that biases and variability differences in time-to-contact judgements reflect only biases and variability differences in speed estimation. Studying to what extent biases and variability differences in online information acquired while viewing a target influence the way we extrapolate its further motion will help us better understand the predictive mechanisms at play

not only when the target is occluded or the observer averts their gaze from the target but also when timing interceptive actions accurately despite unavoidable neural delays [36]. In the present study, we therefore investigate how biases in speed estimation elicited by visual self-motion impact the prediction of object motion.

More specifically, we will test three interconnected hypotheses:

- Predictions of where a moving object will be in the future will be biased (Hypothesis 1a) and more variable (Hypothesis 1b) in the presence of visually simulated self-motion

- Object speed judgements will be biased (Hypothesis 2a, which constitutes a replication of an earlier result of Jörges & Harris, 2021) and more variable (Hypothesis 2b, a more highly powered follow-up to a hypothesis for which Jörges & Harris, 2021 did not find significant support) in the presence of visually simulated self-motion

- The effect of visually simulated self-motion on motion extrapolation can be predicted from its effect on speed estimation, both in terms of bias (Hypothesis 3a) and its variability (Hypothesis 3b)

## Methods

### Apparatus

We programmed all stimuli in Unity 2020.3.30f1. Given the on-going COVID-19 pandemic, some participants who are owners of head-mounted VR devices (HMDs) are tested in the safety of their home. Our experiment is compatible with all major modern HMDs. To minimize variability introduced by the use of different HMDs, each program we use to present stimuli limits both the horizontal and the vertical field of view to 80˚. We don't expect there to be relevant differences in frame rate, as Unity caps the frame rate at 60 Hz. Our programs are, further, much less demanding in processing power than any VR application remote participants are likely to run; that is, frame rate dropping below 60 Hz should occur almost never. If it is safely possible and an ethics approval is granted, we might also test some participants in person in our laboratory. For these participants, we will use an VIVE Pro Eye.

### Participants and recruitment

We recruit participants with HMDs in their possession online for them to perform the experiment in their homes. Recruitment occurs through social media (such as Twitter, Facebook, and Reddit). Some remote participants might also be recruited through the professional recruitment service XpertVR. Since recruiting participants with VR equipment at home is not trivial, we might also rely on York University participant pools to recruit participants for in-person testing. In this case, all applicable guidelines for safe face-to-face testing are fulfilled and exceeded. Participants receive a monetary compensation of 45 CAD for participation in the experiment; participants recruited through the York University participant pools may receive course credit instead of a monetary compensation. All participants are screened for stereo-blindness with a custom Unity program (downloadable on GitHub: https://github.com/b-jorges/Stereotest) in which participants have to distinguish the relative depth of two objects that are matched in retinal size. Participants are only included if they answer correctly on 16 out of 20 trials. The simulated disparity is 200 arcsec. While this allows only for a coarse assessment of the participants' stereovision, our experiment is not critically dependent on a high stereoacuity. We test 20 men and 20 women (see Power Analysis). The experiment was approved by the local ethics committee and is conducted in accordance with the Declaration of Helsinki.

## Stimulus

Each participant performs two main tasks in an immersive VR environment: a prediction task and a speed estimation task. Every participant completes both tasks, and we counterbalance the order in which they complete them such that 20 participants (10 women and 10 men) start with the prediction task, and 20 participants (10 women and 10 men) start with the speed estimation task. All programs we used to present the stimuli are available on Open Science Foundation (https://osf.io/gakp5/), and the Unity projects can be downloaded on Open Science Foundation as well (https://osf.io/6mz4w/).

For both tasks, we display a circle in the middle of the observer's field of view that moves with their head rotation in front of the Earth-stationary simulated world. Participants are instructed to keep this circle surrounding the fixation cross. When the center of the circle is with 2.5˚ (vertically and horizontally) of the fixation cross, it disappears to indicate to the participant that their head is positioned correctly. We further record head position whenever the ball or the ball cloud (see below) are visible. Since recording head position on each frame might slow down the program on older systems, we opt to record the mean rotation (horizontally and vertically) over bins of five frames.

**Prediction.** In the prediction task (see Fig 2A, and see also this video on OSF (https://osf.io/rkg23/), we first show participants a ball of 0.4 m diameter moving laterally 8m in front of them at one out of three speeds (4, 5, 6 m/s). We have used this range of speeds in our previous study [30], while the size of the ball was diminished slightly in comparison to this study (see description of the speed estimation task for the rationale). The ball can travel to the left or to the right. It appears to the left of the observer when it travels to the right, and on the right of the observer when it travels to the left such that the visible part of the trajectory is centered in

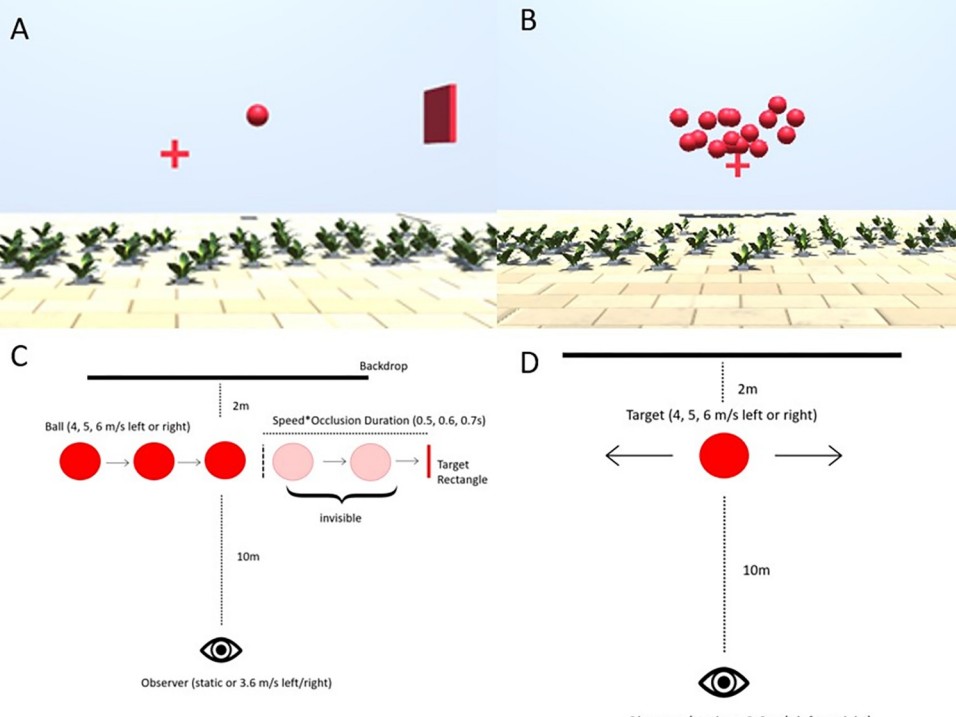

**Fig 2. A**. Screenshot from the prediction task while the ball was visible. **B**. Screenshot from the speed estimation task while the ball cloud was presented. **C**. Schematic of the prediction task. **D**. Schematic of the speed estimation task.

front of the observer. At the same time, a target rectangle is presented on the side towards which the ball is travelling. The ball disappears after 0.5s and participants press the space bar on their keyboard in the moment they think the ball would hit the target. In order to curtail biases in speed perception due to the Aubert-Fleischl phenomenon [37], participants are asked to keep their eyes on a fixation cross that is presented straight ahead of them and slightly below the stimulus (see Fig 2A) and moves with the observer when they experience visual self-motion. The target is presented at a distance that depends on the speed of the ball and the occlusion duration, which can be 0.5 s, 0.6 s, or 0.7 s. Speeds and occlusion durations are chosen such that, when the participant keeps their gaze on the fixation cross, the whole trajectory (including the invisible part) unfolds within a field of view of 60˚, which is well within the effective field of view of any modern HMD. The distance between the point where the ball disappears (the "point of disappearance"; see Fig 2C) and the target rectangle is given by the following equation:

$$Distance = Duration_{Occlusion} * Speed_{Ball} \qquad [1]$$

While the target is visible, participants experience lateral visual self-motion either in the same direction as the ball or in the opposite direction as the ball, or they remain stationary. The self-motion speed ramps up in a Gaussian fashion over the first 50 ms until it reaches 4 m/s, then remains constant for 400 ms, and finally ramps down again over the last 50 ms before the ball becomes invisible. Overall, the observer moves 1.8 m over 500 ms. Please note that the different motion profiles elicit very different retinal speeds: observer motion in the opposite direction of the ball elicits higher retinal speeds overall than for a static observer or for observer motion in the same direction as the ball. Table 1 displays the mean absolute retinal speeds across the trajectory for all conditions. While our previous results [30] suggest that the role of the retinal speeds for the overall precision in motion estimation is subordinate to other sources of variability, retinal speeds are highly correlated with the expected effect of our self-motion manipulation on variability.

We further add a range of occlusion durations (0.1s, 0.2s, 0.3s, 0.4s, 0.8s, 0.9s, 1s) while the observer is static to get an estimate of how variability changes in response to different occlusion durations. Overall, participants complete 225 trials (3 ball speeds * 3 self-motion profiles * 3 occlusion durations * 5 repetitions + 3 ball speeds * 6 occlusion durations * 5 repetitions), which takes around 10 minutes.

Participants complete a brief training of 18 trials before starting the main experiment (see this video on OSF: https://osf.io/4js5w/). The ball travels at one of three speeds (2.5, 3.5, 4.5 m/s), going either left or right, and three occlusion durations (0.45, 0.55, 0.65 s). In the training, the ball reappears upon pressing the spacebar in the position it would have been in at that moment. No visual self-motion is simulated in the training. This allows participants to estimate their error (spatially) and helps them familiarize themselves with the task and the environment.

**Table 1. Mean absolute retinal speeds across the visible part of the trajectory for each combination of ball speed and observer motion profile.** The script in which we derive these values can be found on GitHub (https://github.com/b-jorges/Predicting-while-Moving/blob/main/Geometry%20Prediction.R).

|  | Ball Speed | | |
| --- | --- | --- | --- |
|  | **4 m/s** | **5 m/s** | **6 m/s** |
| Observer Static | 0.4˚/s | 28.4˚/s | 34.0˚/s |
| Same Direction | 22.2˚/s | 5.9˚/s | 11.6˚/s |
| Opposite Directions | 44.4˚/s | 50.4˚/s | 55.8˚/s |

**Speed estimation.**   In the speed estimation task (video on OSF: https://osf.io/xqkgy/), participants are presented with two motion intervals and have to judge which of them is faster. In one interval, they view a ball travelling to the left or to the right. As for the prediction task, this ball can have one of three speeds (4, 5, 6 m/s), and the participant can also experience visual self-motion in the same direction or in the opposite direction or remain static (see Fig 2D). Except for the self-motion intervals, the scene is static just like in the prediction experiment, and object motion occurs relative to the scene. The second motion consists of a ball cloud of 2.5 m width and 1 m height at the same distance to the observer (see Fig 2B). Each ball in this cloud has the same diameter as the main target (0.4m) and balls are generated at one side of the cloud and then move to the other side where they disappear. In our previous study [30], we used smaller balls for the ball cloud, which may have been a factor their judgements of speed being consistently overestimated relative to the single ball. We therefore decided to use the same ball size for both the single ball and the elements of the ball cloud. At any given moment, between 8 and 12 balls are visible. All the balls in the ball cloud move at the same speed and are visible either until they reach the opposite side of the cloud area or until the motion interval ends after 0.5s observation time. The speed of these balls is constant throughout each trial and is governed by PEST staircase rules [38]. For each condition, we employ two pests: one starts 30% above the speed of the single ball from the other motion interval, while the other starts 30% below. The initial step size is 0.6 m/s and each pest terminates either after 37 trials, or when the participant has completed at least 30 trials and the step size drops below 0.03 m/s. We modified the original PEST rules such that the step size is always twice the initial step size (that is, 1.2 m/s) for the first ten trials in order to spread out the values presented to the observer and allow for more robust JND estimates. We limit the range of speeds the ball cloud can take to between one third of the speed of the single ball and three times the speed of the single ball. Participants are asked to maintain fixation on the fixation cross that had the same characteristics as in the prediction task, that is, it is always presented slightly below the stimulus and it moves with the participant as they experience visual self-motion. To keep the visual input identical across both tasks, the target rectangle from the prediction task, while irrelevant for the speed estimation task itself, is present in this task as well. Overall, participants perform between 30 and 37 trials in 18 staircases (two start values, three speeds and three motion profiles) for a total of between 540 and 666 trials.

Before proceeding to the main tasks, participants complete a training session. This training session consists of one PEST of reduced length (between 20 and 27 trials) that starts 30% above the speed of the ball (3 m/s). Participants need to achieve a final step size of below 0.3; otherwise, they are asked to repeat the training. If they fail the training a second time, we exclude them from the analysis. The participants do not experience visually simulated self-motion in this training. This task–including the training–takes about 40 minutes to complete.

Participants can choose to receive the instructions as PDF (can be downloaded from GitHub: https://github.com/b-jorges/Predicting-while-Moving/blob/main/Instructions%20Predicting%20while%20moving.pdf) or watch a video (which can be viewed on YouTube: https://youtu.be/qHTWVyjn0QI and https://youtu.be/JyOZ-duRGmU, respectively).

## Modelling the predictions

To obtain specific predictions corresponding to each hypothesis, we built models of the underlying perceptual processes for both the prediction and the speed estimation task. The instantiation of the model for the prediction task can be found here (on GitHub: https://github.com/b-jorges/Predicting-while-Moving/blob/main/Analysis%20Prediction.R), and the instantiation of the speed estimation model can be found here (on GitHub: https://github.com/b-jorges/

Predicting-while-Moving/blob/main/Analysis%20Speed%20Estimation.R). The implementation of the model that relates performance in both tasks can be found here (on GitHub: https://github.com/b-jorges/Predicting-while-Moving/blob/main/Predictions%20Correlations.R). While a detailed discussion of these models can be found in S1 Appendix, the most important assumptions are the following. The first assumptions reflect our hypotheses:

- The speed of the ball is overestimated by 20% of the presented self-motion speed when observer and ball move in opposite directions [30], with a between-participant standard deviation of 30%. In the prediction task, this overestimation of speed should lead to an underestimation of the time it takes for the ball to travel the occluded distance. No biases are assumed for the Same Directions and Observer Static motion profiles. This assumption reflects Hypothesis 2a.

- While we previously did not find evidence for an impact of visual self-motion on precision [30], we believe that the higher self-motion speeds in this study might enable us to uncover small effects that were not apparent at lower self-motion speeds. The variability in perceived speed is 20% higher for the Opposite Directions motion profile, with a between-participant standard deviation of 30%. No differences in variability are assumed for the Opposite Directions and Static motion profile. This assumption reflects Hypothesis 2b.

- The same effects of self-motion on accuracy and precision are also at play in the prediction task. This assumption reflects Hypotheses 1a and 1b.

- Participants display the same biases in perceived object speed in response to self-motion in the opposite direction of the ball in both the speed estimation and the prediction task. Similarly, variability is impacted equally in both tasks. This assumption reflects Hypotheses 3a and 3b.

We further need to make several assumptions about how the participants process the stimulus independently of the presence of visual self-motion:

- We neglect how exactly participants recover physical speed from angular speed and the perceived depth but to acknowledge the added complexity, we assume a Weber Fraction of 10% for the estimation of the ball speed, which is slightly higher than is generally reported for less complex speed discrimination tasks [39], with a between-participant standard deviation of 1.5%.

- *Prediction task only*: We assume that the computations executed by the visual system are approximated accurately with the physical equation for distance from speed and time (d = v*t), such that the extrapolated time ($t_{extrapolated}$) can be estimated from the distance between the point of disappearance and the target ($d_{perceived}$) and the perceived speed of the ball ($v_{perceived}$):

$$t_{extrapolated} = d_{perceived}/v_{perceived} \tag{2}$$

- *Prediction task only*: The distance between the point of disappearance of the ball and the target rectangle (that is, the occluded distance) is estimated with a Weber Fraction of 5%, as reported in the literature [40]. We further assume that this distance is estimated accurately or, if biases in depth perception impact accuracy, that these biases also impact the perceived speed of the ball in such a way that these biases cancel out. Out of these two scenarios, we

believe the latter is more likely, as it is quite established in the literature that depth is under-estimated in VR [41], and an underestimation of depth would lead to an underestimation of the physical distance between the point of disappearance and the target rectangle. However, this bias in depth perception should also lead the observer to underestimate the physical speed of the ball in the same way, causing both biases to cancel each other out. Between-participant variability is neglected here.

**Prediction.** For the prediction task, under the above assumptions, participants are expected to respond between 0.12 and 0.2s earlier during the Opposite Direction motion profile than during the Static motion profile (Fig 3A). Our model further predicts that visual self-motion during motion observation will lead to higher variability in responses. Measuring the relation between self-motion and variability is not straight-forward because self-motion should cause an underestimation of the occlusion duration. If most noise behaves according to Weber's Law, noise should be proportional to the mean length of the extrapolated interval. A shorter predicted interval should thus in turn be related to lower variability (in absolute terms) even if self-motion has no direct effect on precision. Fig 3B illustrates the expected relationship between biases in prediction, the motion profile, and variability in responses.

**Speed estimation.** Here, we expect to replicate the findings from our previous study [30]: There we found that participants largely estimated speed with the same degree of accuracy when they were static as when they were moving in the same direction as the target. In line with these results, visually simulated self-motion in the opposite direction to the ball should lead to an overestimation of ball speed (Hypothesis 2a; see Fig 4A). Since we use a higher self-motion speed than in our previous study, we also expect that precision will be lower for visual self-motion in the opposite direction to the ball (Hypothesis 2b, see Fig 4B).

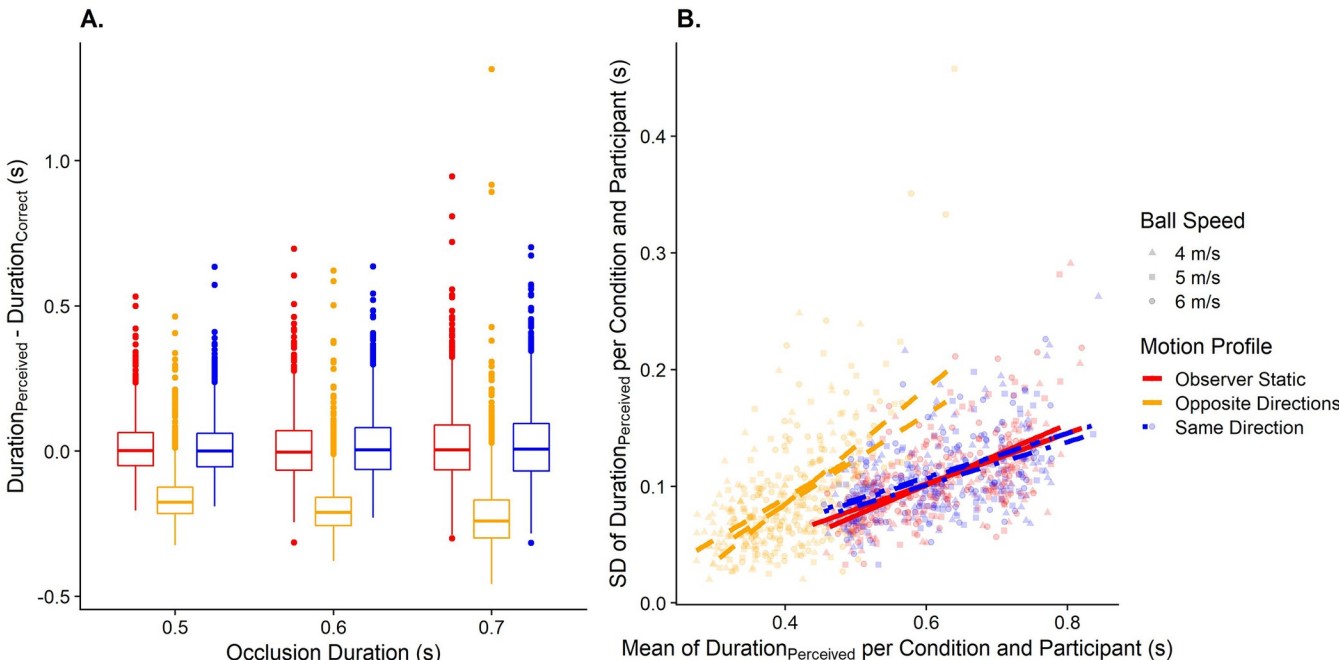

**Fig 3.** A. Predicted data for the timing error in the prediction task, divided up by occlusion durations (x axis) and motion profile (color-coded; left-most: "Observer Static"; in the middle: "Opposite Directions"; right-most "Same Directions"). B. Predicted data for variability in the prediction task. The y axis displays the standard deviation of the extrapolated duration per condition and participant, while the x axis corresponds to the mean of the extrapolated duration per condition and participant. The motion profile is coded with different colors and line types (red and continuous for "Observer Static", yellow and dashed for "Opposite Directions" and blue and dashed-and-dotted for "Same Direction"). The lines are regression lines for their respective condition.

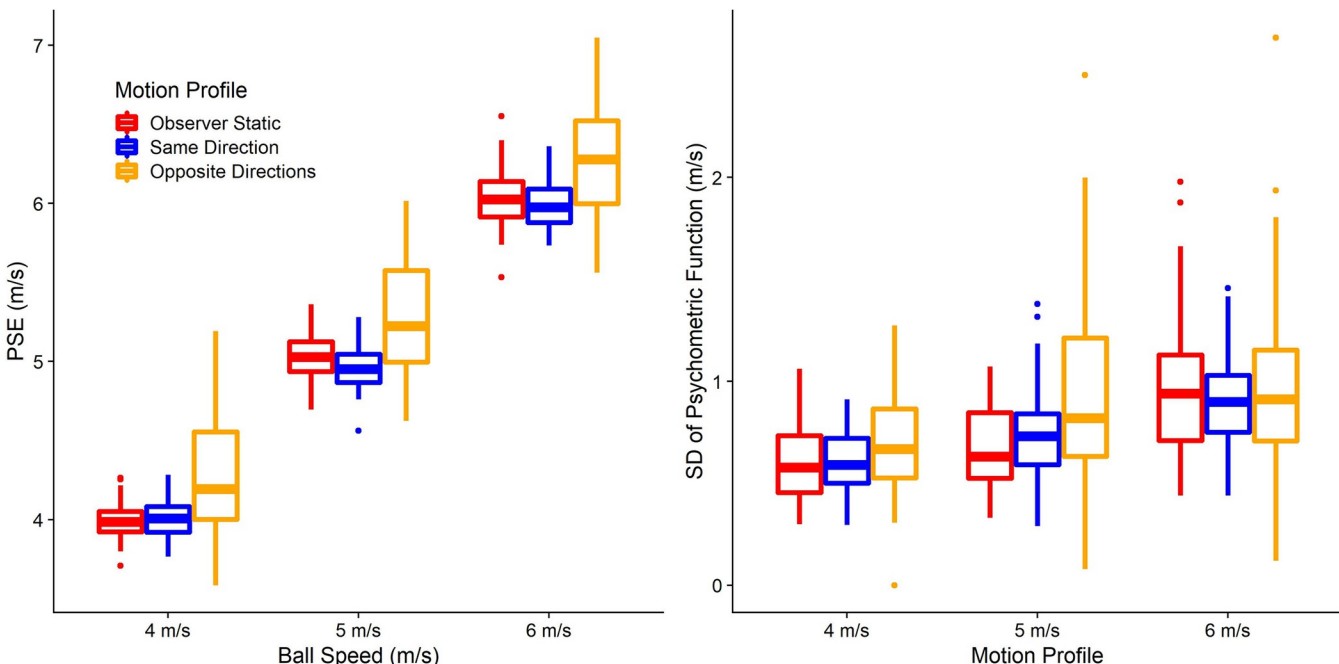

**Fig 4.** A. Predicted PSEs (y axis) for each ball speed (x axis) and motion profile (color-coded; left-most: "Observer Static"; in the middle: "Opposite Directions"; right-most "Same Directions"). B. As A. but for the predicted JNDs.

**A link between speed estimation and predicted time-to-contact.** We further expect the errors observed in the prediction task in response to self-motion to correlate with the errors in the speed estimation task in response to self-motion, indicating that performance in speed perception translate to errors in predicted time-to-contact, both in terms of accuracy (Fig 5A) and precision (Fig 5B).

## Data analysis plan

We first perform an outlier analysis. For the prediction task, we exclude all trials where the response timing was more than three times the occlusion duration, which indicates that the participant has not paid attention and missed the trial. For the speed estimation task, we exclude participants where more than 20% of presented ball cloud speeds were at the limits we set for the staircase (one third of the speed of the single ball and three times the speed of the single ball). For all analyses related to precision, we further exclude all conditions where we obtained a standard deviation of 0.01 or lower. According to our simulations, this should occur very rarely, and taking the log of such low values, as we do for the precision analyses to counteract the expected skew in these distributions, would lead to extremely small numbers that could bias results unduly. We also remove all trials where the head rotation was outside of the permitted range (+- 2.5˚) for half or more of the recorded bins.

Unless noted otherwise, we compute bootstrapped 95% confidence intervals as implemented in the confint() function for base R [42] to determine statistical significance.

**Prediction.** To test Hypotheses 1a regarding accuracy, we use Linear Mixed Modelling as implemented in the lme4 package [43] for R. The corresponding script can be found here (on GitHub: https://github.com/b-jorges/Predicting-while-Moving/blob/main/Analysis%20Prediction.R). We fit a model with the temporal error as dependent variable, the motion profile ("Observer Static", "Same Direction" and "Opposite Directions") as fixed effect, and

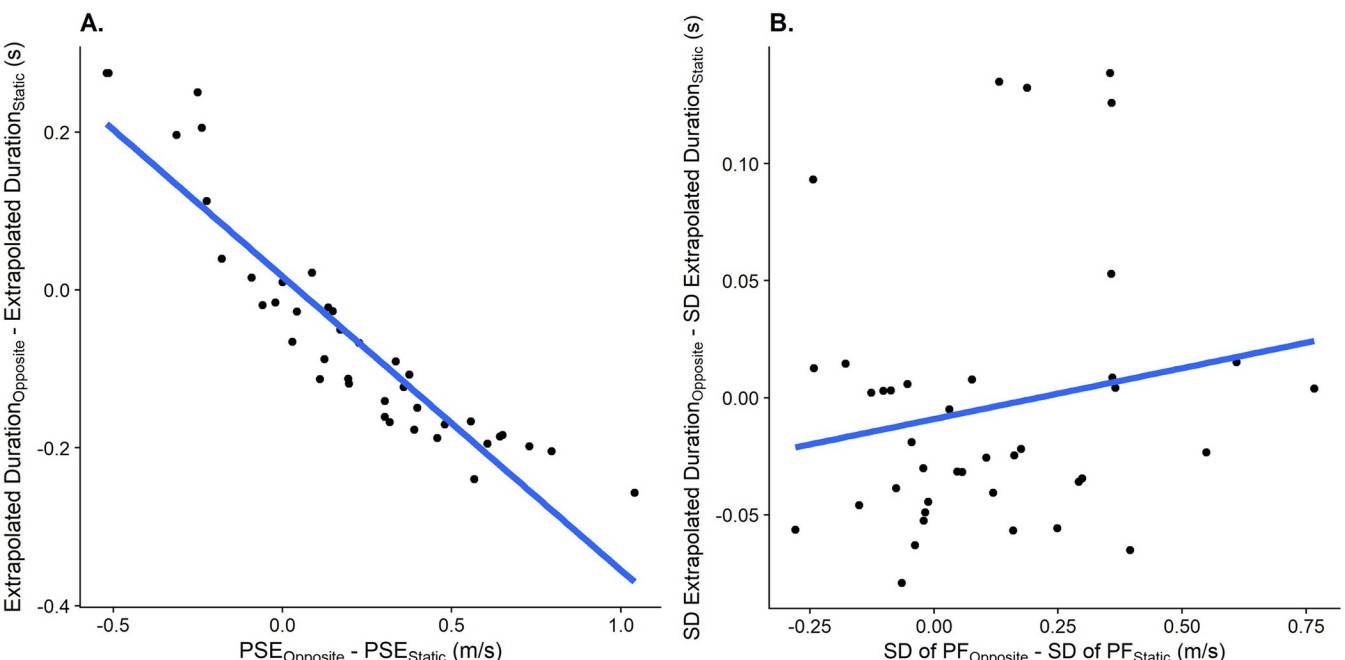

**Fig 5.** A. Relationship between the difference in PSEs between the Opposite Directions motion profile and the Observer Static motion profile in the speed estimation task (x axis) and the difference in predicted durations between these motion profiles (y axis). One data point corresponds to one participant. B. As A., but for the relation between the JND differences in the speed estimation task between the "Opposite Directions" motion profile and the "Observer Static" motion profile and the differences in standard deviations between these motion profiles.

random intercepts and random slopes for the speed of the ball per participant, as well as random intercepts for the occlusion duration as random effects. In Wilkinson & Rogers notation (1973) [44], this model reads as follows:

$$Error \sim Motion\ Profile + (Speed_{Ball}|Participant) + (1|Occlusion\ Duration) \qquad (3)$$

We expect the regression coefficient corresponding to the motion profile "Opposite Directions" to be negative and significantly different from zero.

For Hypothesis 1b regarding precision, we need to take into account one possible confound: differences in timing accuracy can impact variability: overestimating the time it takes the ball to hit the rectangle could be connected to a higher variability, while underestimating the time could lead to lower variability. For this reason, we first compute the means and standard deviations of extrapolated durations for each condition and participant. We then fit a test model with the standard deviations as dependent variable, the mean timing error and the motion profile as fixed effects, and random intercepts as well as random slopes for ball speeds per participant and random intercepts for the occlusion durations as random effects:

$$\log(SD\ of\ Extrapolated\ Time) \\ \sim Mean\ of\ Extrapolated\ Time + Motion\ Profile + (Speed_{Ball}|Participant) + (1|Occlusion\ Duration) (4)$$

We further fit a null model without the motion profile as fixed effect:

$$\log (SD\ of\ Extrapolated\ Time) \\ \sim Mean\ of\ Extrapolated\ Time + (Speed_{Ball}|Participant) + (1|Occlusion\ Duration) \qquad (5)$$

We then compare both models by means of a Likelihood Ratio Test to determine whether the motion profile explains significantly more variability than the test model which already

takes into account biases in extrapolated time. We will not interpret the regression coefficients as means and standard deviations are likely to be correlated, which may lead to biased regression coefficients.

**Speed estimation.** To test Hypotheses 2a and 2b (script can be found here, on GitHub: https://github.com/b-jorges/Predicting-while-Moving/blob/main/Analysis%20Speed%20Estimation.R) regarding speed estimation, we first use the R package quickpsy [45] to fit psychometric functions to the speed estimation data, separately for each participant, speed and motion profile. Quickpsy fits cumulative Gaussian functions to the data by direct likelihood maximization. The means of the cumulative Gaussians correspond to the Points of Subjective Equality (PSEs) and their standard deviations correspond to the 84.1% Just Noticeable Differences (JNDs).

To assess whether the motion profile biased the perceived speed significantly, we fit a Linear Mixed Model with the PSEs as dependent variable, the self-motion profile as fixed effect, and random intercepts and random slopes for the ball speed per participant as random effects:

$$PSE \sim Motion\ Profile + (Speed_{Ball}|Participant) \tag{6}$$

We expect that the regression coefficient for the motion profile "Opposite Directions" will be positive and significantly different from zero, indicating that speed is overestimated when observer and target move in opposite directions as compared to when the observer is static.

Regarding precision, the same considerations apply as for the prediction task: in addition to a direct effect of the self-motion profile, biases elicited by the different self-motion profiles can impact precision. For this reason, we use a model comparison-based approach similar to the one used above. Separately for the "Same Direction" and "Opposite Directions" motion profiles, we first fit a test model that contains the log JNDs as dependent variable, the self-motion profile and the PSEs as fixed effects, and random intercepts as well as random slopes for ball speed per participant as random effects.

$$log(JND) \sim Motion\ Profile + PSE + (Speed_{Ball}|Participant) \tag{7}$$

We also fit a null model without the motion profile as fixed effect:

$$log(JND) \sim PSE + (Speed_{Ball}|Participant) \tag{8}$$

Finally, we compare both models with a Likelihood Ratio Test and we expect the test model (Eq 7) to be a significantly better fit than the null model (Eq 8). As for the prediction task, we will not interpret the regression coefficients obtained in this analysis.

**A link between speed estimation and prediction.** To test Hypotheses 3a and 3b (script can be found here, on GitHub: https://github.com/b-jorges/Predicting-while-Moving/blob/main/Analysis%20Correlation.R), we first prepare the prediction data by computing means and standard deviations of the extrapolated time for each participant. We then calculate the difference in performance (mean and standard deviations for the prediction task and PSEs and JNDs for the speed estimation task) between the "Opposite Directions" motion profile and the "Observer Static" motion profile for both tasks for each participant.

For accuracy, we then determine to what extent PSE differences between the Opposite Direction motion profile and the Observer Static motion profile obtained in the speed estimation task predict the mean extrapolated time in the prediction task. For this purpose, we fit a Linear Model with the difference in mean motion extrapolation errors between the motion profiles as the dependent variable and the difference in PSEs between the motion profiles as

the independent variable:

$$Mean\ Difference\ in\ Extrapolated\ Time - to - Contact\ (Opposite - Static)$$
$$\sim PSE\ Difference\ (Opposite - Static) \tag{9}$$

We expect that the regression coefficient for the fixed effect "PSE Difference (Opposite–Static)" will be negative and significantly different from zero, indicating that a stronger effect of self-motion on PSEs is linked to a larger effect of self-motion on the estimated time-to-contact.

For precision, the same complication as for Hypothesis 1b applies: A correlation between the effect of visual self-motion on the precision of speed estimation and on the precision of the predicted times-to-contact could be due to biases introduced by visual self-motion. If visual self-motion in the opposite direction, for example, leads to too-early responses, the extrapolated intervals become shorter. A shorter interval, in turn, would lead to higher precision. Therefore, to test whether the difference in precision observed in the speed estimation task was significantly related to the variability in the prediction task even after accounting for biases, we need to determine whether the effect of visual self-motion on JNDs predicts any variability beyond the variability that is already explained by the bias in motion extrapolation. To test this hypothesis, we first fit a test model with the variability difference between the "Opposite Directions" motion profile and the "Observer Static" motion profile in the prediction task as the dependent variable and the mean difference between these motion profiles and the difference in JNDs in the speed estimation task as the independent variables (as a measure of bias introduced by visual self-motion):

$$Variability\ Difference\ in\ Extrapolated\ Time\ (Opposite - Static)$$
$$\sim Mean\ Difference\ in\ Extrapolated\ Time\ (Opposite - Static) + JND\ Difference\ (Opposite - Static) \tag{10}$$

We also fit a null model without the JND difference as independent variable:

$$Variability\ Difference\ in\ Extrapolated\ Time\ (Opposite - Static)$$
$$\sim Mean\ Difference\ in\ Extrapolated\ Time\ (Opposite - Static) \tag{11}$$

Then, we use a Likelihood Ratio Test to determine whether the test model (with the JND difference as fixed effect) was significantly better than the null model. We expect the test model (Eq 10) to be a significantly better fit than the null model (Eq 11). As above, the regression coefficient will not be interpreted.

**Model fitting.** These analyses only serve to demonstrate that performance in both tasks is related, but they don't provide insight into how strongly they are related. We will therefore use the models outlined in the section "Modelling the Predictions", and described more in detail in S1 Appendix, to fit parameters that capture biases and precision differences in perceived speed due to self-motion (which we had set to 20% on average to model our predictions and conduct the power analyses) in both tasks separately.

We will use a two-step approach to fit these parameters: since we expect biases to affect variability in performance but do not expect variability to affect biases, we first set the variability parameter (capturing the impact of self-motion on the variability of perceived speed) to zero and fit the accuracy parameter (capturing the impact of self-motion on mean perceived speed). To do so, we minimize the root median squared error between the observed mean difference in timing errors between baseline and the self-motion condition and the difference in timing error in the simulated dataset across all speeds and occlusion durations. We perform this optimization by using the Brent method [46] as implemented in the optimize function in

base R. In the second step, we set the accuracy parameter for each participant to the one fitted in the first step and fit the precision parameter. Here, we minimize the root median squared error between the difference between the standard deviation of the observed difference in timing errors in the baseline condition and the self-motion condition and the respective simulated values. We use the same approach to obtain these parameters for the speed estimation task as well but use the observed and simulated PSEs (for accuracy) and JNDs (for precision).

Once we have obtained one accuracy parameter and one precision parameter for each participant and task, we perform a simple linear regression between the parameters fitted for the prediction task and the speed estimation task (separately for accuracy, see Eq 12, and precision, see Eq 13) to determine to what extent performance in one task is indicative of performance in the other:

$$Bias_{Prediction} \sim Bias_{Speed\ Estimation} \tag{12}$$

$$Precision\ Difference_{Prediction} \sim Precision\ Difference_{Speed\ Estmation} \tag{13}$$

Since these parameters are scaled the same way for both tasks (the effect of self-motion on accuracy/precision as a fraction of presented self-motion speed), we expect regression coefficients of around 1 in both analyses. A value of above 1 would mean that the effect of self-motion is stronger in the prediction task than in the speed estimation task, and vice-versa. To test this prediction we compute 95% confidence intervals which we expect to contain a value of 1.

**An effect of visual self-motion in the same direction as the ball.** While our earlier results [30] suggest that visual self-motion in the same direction as the observer should not have any effect on perceived speed, we perform all analyses outlined in this section equivalently for the "Same Direction" motion profile as well.

## Power analysis

Since power for complex hierarchical designs cannot be computed analytically, we used Monte Carlo simulations to determine power for all statistical tests outlined in the previous section: we used our models for the prediction task and the speed estimation task to first simulate full datasets. Then, we performed the analyses detailed above over each of these simulated datasets and determined the results for each combination of number of participants and number of trials. To keep the computational cost manageable, we used the faster, but more bias-prone Satterthwaite approximation, as implemented in the lmerTest package [47] for R, to assess statistical significance rather than bootstrapped confidence intervals. The script used for the power analyses can be found here (on GitHub: https://github.com/b-jorges/Predicting-while-Moving/blob/main/Power%20Analysis.R).

We repeated this process 250 times for all combinations of 20, 30 and 40 participants, 5, 9 and 13 repetitions per condition in the prediction task, and 20 to 27, 30 to 37, and 40 to 47 trials per pest, which makes for an average of 50, 70 and 90 trials per condition, respectively, for the speed estimation task. The results are shown in Fig 6. For the precision in the prediction task, using 9 repetitions per condition appears to add a considerable amount more power than using only 5 repetitions, while the added benefit of another 4 repetitions for a total of 13 is small. However, the prediction task is very quick to do, taking only around 10 minutes even with 13 repetitions per condition. Similarly, 70 trials per condition increases the power to detect an effect on precision significantly more than using only 50 trials, while the added benefit of 90 trials is marginal. Since the speed estimation task takes much longer and is more

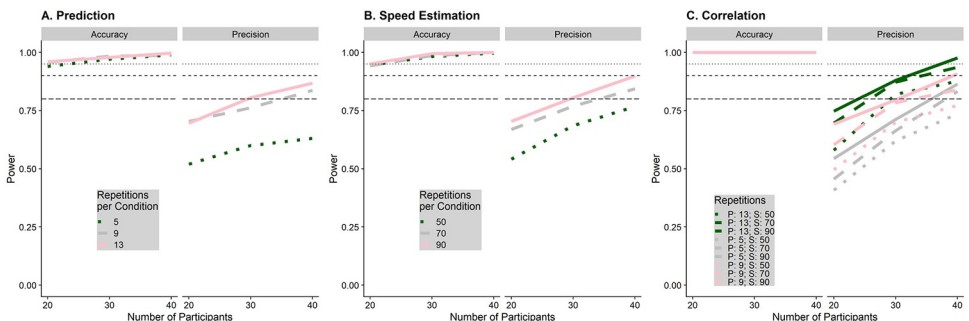

**Fig 6.** Simulated power for the prediction task (A), the speed estimation task (B) and the correlation between performance in speed estimation and speed prediction (C), separately for the statistical tests referring to biases (accuracy) and variability (precision). The number of participants for which we simulated power is on the x axis, while the number of trials for each task is coded with different shades of green and line types. The horizontal lines indicate a power level of 0.8, 0.9 and 0.95 respectively.

fatiguing than the prediction task, we judge this marginal increase in power to be not worth the additional time spent by the participant. We thus opt for a combination of 40 participants, 13 repetitions per condition in the prediction task, and 70 trials per condition in the speed estimation task, which allows us to achieve a power of at least 0.85 for all statistical tests.

We also used our power analyses to determine that all of our statistical analyses led to an expected false positive rate of 0.05 in absence of a true effect.

## Supporting information

**S1 Appendix.**
(DOCX)

## Author Contributions

**Conceptualization:** Björn Jörges, Laurence R. Harris.

**Data curation:** Björn Jörges.

**Formal analysis:** Björn Jörges.

**Funding acquisition:** Laurence R. Harris.

**Investigation:** Björn Jörges.

**Methodology:** Björn Jörges.

**Project administration:** Björn Jörges.

**Resources:** Björn Jörges.

**Software:** Björn Jörges.

**Supervision:** Laurence R. Harris.

**Visualization:** Björn Jörges.

**Writing – original draft:** Björn Jörges.

**Writing – review & editing:** Laurence R. Harris.

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
