## [Decision Letter · Decision Letter 0]

13 Feb 2022

PONE-D-21-37478The Impact of Visually Simulated Self-Motion on predicting object Motion – A Registered Report ProtocolPLOS ONE

Dear Dr. Jörges,

Thank you for submitting your Registered Report Protocol to PLOS ONE. After careful consideration, we feel that it has merit but requires some revision to meet PLOS ONE’s publication criteria as it currently stands. Therefore, we invite you to submit a revised version of the manuscript that addresses the points raised during the review process. All of the reviewers share my enthusiasm for the potential of your study, and each made reasonable requests, suggests, and queries to help improve it so it will be suitable for acceptance upon completion. (Please note some of the auto-populated text provided by PLOS ONE might sound strange, as much of it is more suitable for a normal manuscript. It was clear to me and the reviewers what we were assessing here!)

We look forward to receiving your revised manuscript.

Kind regards,

Michael J Proulx, Ph.D.

Academic Editor

PLOS ONE

Journal Requirements:

“BJ and LRH are supported by the Canadian Space Agency (CSA) (CSA: 15ILSRA1-York). LRH is supported by a Discovery Grant from the Natural Sciences and Engineering Research Council (NSERC) of Canada (NSERC: RGPIN-2020-06093). The funders did not play any role in the study design, data collection and analysis, decision to publish, or preparation of the manuscript.”

“BJ and LRH are supported by the Canadian Space Agency (CSA) (CSA: 15ILSRA1-York). LRH is supported by a Discovery Grant from the Natural Sciences and Engineering Research Council (NSERC) of Canada (NSERC: RGPIN-2020-06093). The funders did not play any role in the study design, data collection and analysis, decision to publish, or preparation of the manuscript.”

Reviewers' comments:

Reviewer's Responses to Questions

**Comments to the Author**

1. Does the manuscript provide a valid rationale for the proposed study, with clearly identified and justified research questions?

Reviewer #1: Partly

Reviewer #2: Yes

Reviewer #3: Yes

2. Is the protocol technically sound and planned in a manner that will lead to a meaningful outcome and allow testing the stated hypotheses?

Reviewer #1: Yes

Reviewer #2: Partly

Reviewer #3: Yes

3. Is the methodology feasible and described in sufficient detail to allow the work to be replicable?

Reviewer #1: Yes

Reviewer #2: Yes

Reviewer #3: Yes

4. Have the authors described where all data underlying the findings will be made available when the study is complete?

Reviewer #1: Yes

Reviewer #2: Yes

Reviewer #3: Yes

5. Is the manuscript presented in an intelligible fashion and written in standard English?

Reviewer #1: Yes

Reviewer #2: Yes

Reviewer #3: Yes

6. Review Comments to the Author

You may also provide optional suggestions and comments to authors that they might find helpful in planning their study.

Reviewer #1: The proposed experiment is solid. It is however not set up clearly in the introduction: why does this need to be done, what gap in the literature does it fill? It is unclear to me what we learn from this effort.

For the rest, I have minor comments:

First few sentences of abstract velocity and speed are both used intermixed, to my understanding they do not have the same meaning. Stick to velocity until you get specific about speed.

P2, lines 13-18, the modelling results by Layton & Niehorster, https://doi.org/10.1371/journal.pcbi.1007397, is of relevance here. Further relevant to this page is https://doi.org/10.1177/2041669517708206.

P2, line 32: is there a “we” missing in this sentence about commitment? More generally, I am wondering why this complicated paragraph is needed at all, or at least it could be built up differently. It has been shown previously that for static observers of simulated self-motion, flow parsing is incomplete, i.e., not all of the self-motion is removed from the retinal motion of the object when judging object motion (see e.g. several warren & Rushton work, Niehorster & Li, 2017). You can then choose to speculate on some reasons, but label them as such from the beginning, the start of the story is that flow parsing is incomplete. That makes this easier to read and sets up the story more directly.

P2-3: I think reference should be made to the work of Wertheim, e.g.

Wertheim, A. H. (1994). Motion perception during self motion: The direct versus inferential controversy revisited. Behavioral and Brain Sciences, 17(2), 293–311. https://doi.org/10.1017/S0140525X00034646

Wertheim, A. H. (2008). Perceiving motion: Relativity, illusions and the nature of perception. Netherlands Journal of Psychology, 64(3), 119–125. https://doi.org/10.1007/BF03076414

He has theories and experimental work about how visual reference signals indicating the velocity of the eyeballs in space affect perception of motions in that space, both in terms of threshold and noise. That seems very directly related here, and if I remember and understand his theory correctly, it yields the additional prediction that the opposite motion condition will yield less overestimation than the same motion direction conditions yields underestimation, because a subtractive JND comes into play in both cases.

The arrows in figure 1 are unclear to me, why is noise increasing three times along the flow parsing pathway?

I’m struggling a little bit with the word predicted. At least, it is not motion that is being predicted, it is time to contact that is being predicted. The motion is known, albeit presumably misperceived, and supposedly held constant to yield the time to contact estimate. This held constant assumption is critical for your analysis. This is sloppy around the end of page 3, for instance, but occurs throughout. Perhaps the term extrapolation fits better here?

Relatedly, p4, line 4: remove the word motion from motion prediction, then the sentence makes sense to me.

P4 line 6, estimates, judgments may be a more appropriate word? Or percepts?

P4, line 25 and other links: please print the actual link in the article text. This way I is preserved better, e.g. in case the article is printed.

P4, line 35: “you”: rewrite

P5: line 6-7: rather critical sentence is incomplete, what happens after occlusion duration?

P5: why a gaussian velocity profile and not just a constant? This seems to me to complicate the situation. Assuming incomplete flow parsing, the object to be judged will also be seen to accelerate and decelerate (or vice versa) in the two movement conditions. What object speed is then used for the judgment, some kind of (weighted) average?

Do participants receive instructions about head motion? What happens to their view of the virtual world when they move their head, does it counterrotate so that the virtual environment is perceived as rigid? Would head movement make your data harder to analyze / add additional unwanted variability to your study (e.g. it could conceivably differ between conditions, although I am not directly aware of studies suggesting that the lateral simulated self-motion will induce head motion).

P5, line 23: is your task doable at very short durations (0.1 and 0.2 s especially)?

I see that the ball casts a shadow on the ground. Is that a deliberate choice? Speed of the shadow over the tiles on the ground (relative speed between the two) is a direct cue to ball speed that could be used more straightforwardly than motion of the ball itself.

Speed estimation: describe more clearly that its only the balls in the ball cloud that move, not the whole scene.

Is 37 trials sufficient for a JND estimate? Simulation work by Prins on his Bayesian staircase suggested you need more like 100 or so, if I recall correctly.

Are participants able to fixate the cross when it is so close to the rapidly moving ball cloud?

Do you experience induced motion in the fixation cross during simulated self-motion? May the opposite direction of this induced motion confound your results?

Do you use the same participants for the two tasks? Results such as Niehorster & Li 2017 suggest there may be wide variation between participants in how complete flow parsing is. That should give you ample variability between participants to do a strong test of correlation for you H3.

P 7, line 18-22. Why you assume that flow parsing is incomplete only in the opposite motion condition, but not in the same motion condition? That needs to be justified. Same for the precision prediction.

P 8, assumption atop the page: and you assume the distance is perceived correctly and is not affected by background motion. Can you justify this? If both perceived v and d may vary between conditions, you have a problem.

Reviewer #2: The study aims to examine an interesting topic: how self-motion influences prediction when interacting with moving objects. The abstract mentions a clear prediction: that perceived object speed is likely to be biased and more variable during self-motion, because separating object motion from self-motion might give rise to systematic errors and must presumably give rise to more variability. The authors propose to study this by presenting virtual moving targets during simulated self-motion (or absence thereof). They examine judgments of target speed by having participants compare the speed of the target with that of a moving cloud of dots within a static window. They examine prediction by having participants press a button when a ball reaches a target. The ball is occluded before it reaches the target. The prediction mentioned in the abstract makes perfect sense, but I feel that it is a very weak prediction: that there will be a correlation between performance in the two tasks. Specifying what one expects to be correlated might change this. I think it is a bit trivial that performance on the two tasks across different speeds of self-motion is correlated, but maybe the authors are referring to correlations across participants within each value of self-motion. Otherwise, maybe it makes more sense to check whether the values are similar, rather than only whether they are correlated (as in de la Malla et al., 2018, Errors in interception can be predicted from errors in perception. Cortex 98, 49-59). I actually see a theoretical complication in interpreting the data. Since the self-motion presumably shifts the goal (the target rectangle) as much as it does the ball, why would you expect any bias in judging self-motion to influence the timing of the tap? I think this needs to be explained.

Another issue that needs justification is the use of a fixation point. Apart from making the task quite unnatural, it also introduces many complications. First of all, how will fixation be ensured. It is very difficult to keep fixating while making judgments about moving targets, and small periods of pursuit at critical moments might influence one’s judgments. Secondly, the participants might make several of the judgments with respect to the fixation point. The fixation point does not move with the simulated self-motion, so its motion relative to the surrounding also needs to be interpreted. It also provides a reference in time for the button press: the time it took the ball to reach/cross fixation. At the very least the authors should explain why they have a fixation point, and how this might influence their results. I would consider not requiring fixation.

When the authors write “the process of flow parsing should add noise and lead to object speed judgements being more variable during self-motion” they are actually making some assumptions. Although these assumptions are probably reasonable, I think the authors should be explicit about the details. Assuming that people use some kind of flow parsing mechanism to separate object motion from self-motion, they presumably also have to do so when there is no self-motion. Thus, the assumption is that speed judgments become more noisy when self-motion is faster, just as they become more noisy (at least in absolute terms; it could be a fixed Weber fraction) when the object moves faster. Being very explicit about the assumptions will help the reader follow the reasoning. It might also be worthwhile more explicitly considering the consequences of the visual self-motion information being in conflict with information from other sources. Following Figure 1, perceived self-motion should be weak because 3 of the 4 ‘senses’ of the multisensory integration indicate that there is no self-motion. There is no evident reason for an asymmetry between motion with or against the ball. I am also not so sure about this interpretation of ‘flow parsing’. Flow parsing refers to the ability to separate object motion from self-motion from the visual information alone. That is indeed necessary for the proposed processing, but I don’t think that a multisensory value of self-motion is normally considered as an input to flow parsing, so maybe the terminology should be adjusted here. Actually, many of the claims and assumptions do not appear to be necessary for answering the question as to whether biases in speed estimation give the anticipated errors in prediction, so probably the introduction (and methods) can be simplified. Moreover, the last pair of hypotheses are what the authors really want to test (I think). They need to check that their manipulation (simulated self-motion) influences judged object speed (and its variability) but actually they already know that it will. Hypothesis 2 is therefore a bit superfluous. They plan to examine whether motion prediction is also influenced (Hypothesis 1) and whether it is influenced in the same manner (hypothesis 3). If it is influenced in the same manner, it must be influenced, so hypothesis 1 is also superfluous. This gives a much clearer study with one hypothesis (with two components: bias and variability).

There are also a number of things to consider in the methods. Especially if people will be tested at home, the authors might want to consider the extent to which participants are allowed to move their heads, and whether such head movements will be compensated for.

Why are stereoblind participants excluded? Do the authors expect their performance to be different? Is it a good idea to always center the trajectory in front of the observer, especially when that position is indicated by a fixation point? Maybe the authors should consider adding some jitter to the position. Otherwise the task could be performed by pressing the button after the same time from when the ball reaches fixation as the time between the ball appearing and it reaching fixation.

It appears to me from the video that the target disappears when the participant presses the button. Is that correct? This should be mentioned explicitly. Since the task is to press the button when the ball would hit the target, this task could be interpreted as judging the time of collision of two moving items, rather than in terms of self-motion. If the target’s motion is underestimated due to motion in the surrounding one might therefore find no effect even though the hypothesis is true. Is there some reason to exclude this possibility? Why was this velocity profile chosen for the self-motion? Not having a constant speed means that the response could be different for the two tasks simply because the moment that is considered relevant is different: for judging speed, presumably only the average speed is relevant, whereas for prediction the change in speed is presumably also relevant. I assume that the training on the prediction task was always with the observer static. This should be specified. Why is there no target in the speed estimation task (in the condition with a single ball)? Might this not influence the comparison? Nice instruction video! I assume you also have a version with the other order.

The status of the assumptions in the predictions section is not quite clear. Some of the assumptions are predictions based on earlier findings, but if the current results turn out to be slightly different it is not a problem. For instance, if the speed of the ball is overestimated by 30% rather than 20% at this speed (or the Weber fraction is not 10%) the reasoning will still hold in the same manner. In the case of the variability it might even be a problem if the results were identical to the previous ones (no influence of self-motion). The third assumption is very philosophical. How would you know whether they have the same bias other than by comparing performance in the two tasks, which is what the study was planned to examine so it cannot be an assumption. The same is true for Equation 3. The Weber fraction of 5% for distance judgments is presumably really an assumption that must be considered when converting speed judgment uncertainty into temporal uncertainty using Equation 3. Maybe explain exactly how this is done and therefore how sensitive the result is to deviations from this value.

In the motion prediction section I think it would be a good idea to clarify that certain predictions are based on earlier research, while others are based on reasoning. This might be important for the interpretation, because not finding the asymmetrical influence of background motion, for instance, need not affect the general conclusion, whereas not finding an increase in the standard deviation with the magnitude would make some of the proposed analysis meaningless.

In Figure 5A I am guessing the y-axis should be in s, not m/s. Why do the authors anticipate precisely this relationship? I think the authors can be a bit more specific about the actual values. Presumably these duration values are obtained by multiplying the difference in PSE by the occlusion time, or something like that. I would be specific, because that is what makes pre-registration a powerful tool. Figure 5B also confused me. If the authors expect such a mess, why bother?

I am not very familiar with the Wilkinson & Rogers notation, so I may be wrong, but it appears from Equation 4 that the authors assume linear, independent effects of observer motion, ball motion and occlusion duration. Why? Would you not for instance expect a larger effect of speed for a longer occlusion duration? Just under that equation the authors speak of biases in timing error. Do they mean systematic errors? This is not really a bias but a potential finding: that observer motion influences timing errors. What would not finding such an effect mean? Maybe the target position is shifted to the same extent as the ball, so their effects cancel? I see many potentially interesting issues to explore, but the idea of pre-registration is to precisely specify what you are testing. For this, I think the authors need to better specify which effect they expect and why. For equations 5 and 6 the measure is clear: all that matters is whether including the Motion Profile in the model provides a significant improvement. Finally, it seems that equation 9 is evaluating whether the judged speed influences the judged timing. Is this really what you want to know? Should you not be testing whether differences in judged speed can fully account for the differences in timing? By equations 10 and 11 you lost me completely. If the time difference and the JND difference are not independent this might give confusing results.

In the power analysis I do not see any measure of the original assumed variability and effect size. Maybe I missed something.

Reviewer #3: Overall, I believe that your study covers an interesting and important topic. It is well designed and the hypotheses are clear and well based on previous literature. I have some suggestions for improvement listed below.

On Page 2, line 21 you mention that “in many virtual reality (VR) applications, vestibular and proprioceptive cues signal that the body is at rest, while the visual optic flow cues simultaneously indicate self-motion.” Could you provide some examples and also explain why this is the case just in some VR applications but not others (e.g. is it due to properties of the hardware or the virtual environment itself?).

On Page 3 lines 11-19 there seem to be references missing for some of the statements you make.

On Page 3, Line 24 you say “it would seem logical that the prediction reflects this bias in motion estimation”. I would like to see a more detailed explanation for this assumption since it is critical for your study. I find that entire paragraph containing explanations that are a bit rushed and unclear.

I greatly appreciate the attention given to the participant sample size and counterbalance of gender and order of conditions.

I am however not sure of the acceptance of any VR HMDs owned by participants. You mention that in-person testing would be conducted on a VIVE Pro Eye if granted permission, but it is expected that participants may possess different HMDs such as Quest 1 or 2, which have significantly different specifications and most importantly, interaction methods (e.g. controllers). Especially for time-sensitive stimuli that you present, it would be important to first conduct a pre-assessment of how your Unity code runs on these HMDs. I understand that due to COVID restrictions currently in place you would have to test remotely, but I believe more should be done to mitigate potential limitations arising from this. Perhaps one option would be to cap the framerate and field of view to certain parameters which are compatible with those HMDs that are lowest in terms of specifications that you would still accept in your study.

For both tasks it is unclear how the speeds and sizes of the stimuli were determined. Was that based on previous literature? if so it should be mentioned or otherwise it should be based on piloting data.

7. PLOS authors have the option to publish the peer review history of their article (what does this mean?). If published, this will include your full peer review and any attached files.

Reviewer #1: No

Reviewer #2: No

Reviewer #3: No

---

## [Author Response · Author response to Decision Letter 0]

30 Mar 2022

Reviewer #1:

The proposed experiment is solid. It is however not set up clearly in the introduction: why does this need to be done, what gap in the literature does it fill? It is unclear to me what we learn from this effort.

(1.1) It is important to study prediction to know how humans deal with (a) occlusions and (b) neural delays in the transduction of motor commands. When studying how humans predict how objects will move in their environment, it is then important to know what information these predictions are based on. While it seems fairly likely that the same biases observed in speed estimation would also apply to the prediction of time-to-contact, we believe that this shouldn’t be assumed, and it has, to our knowledge, not been studied empirically. We have reworked the introduction to make the relevance of the proposed experiment clearer.

First few sentences of abstract velocity and speed are both used intermixed, to my understanding they do not have the same meaning. Stick to velocity until you get specific about speed.

(1.2) Thank you, addressed.

P2, lines 13-18, the modelling results by Layton & Niehorster, https://doi.org/10.1371/journal.pcbi.1007397, is of relevance here. Further relevant to this page is https://doi.org/10.1177/2041669517708206.

(1.3) Thank you, these are indeed highly pertinent additions.

P2, line 32: is there a “we” missing in this sentence about commitment? More generally, I am wondering why this complicated paragraph is needed at all, or at least it could be built up differently. It has been shown previously that for static observers of simulated self-motion, flow parsing is incomplete, i.e., not all of the self-motion is removed from the retinal motion of the object when judging object motion (see e.g. several warren & Rushton work, Niehorster & Li, 2017). You can then choose to speculate on some reasons, but label them as such from the beginning, the start of the story is that flow parsing is incomplete. That makes this easier to read and sets up the story more directly.

(1.4) We agree with the reviewer’s suggestions and have edited this paragraph accordingly.

P2-3: I think reference should be made to the work of Wertheim, e.g.

Wertheim, A. H. (1994). Motion perception during self motion: The direct versus inferential controversy revisited. Behavioral and Brain Sciences, 17(2), 293–311. https://doi.org/10.1017/S0140525X00034646

Wertheim, A. H. (2008). Perceiving motion: Relativity, illusions and the nature of perception. Netherlands Journal of Psychology, 64(3), 119–125. https://doi.org/10.1007/BF03076414

He has theories and experimental work about how visual reference signals indicating the velocity of the eyeballs in space affect perception of motions in that space, both in terms of threshold and noise. That seems very directly related here, and if I remember and understand his theory correctly, it yields the additional prediction that the opposite motion condition will yield less overestimation than the same motion direction conditions yields underestimation, because a subtractive JND comes into play in both cases.

(1.5) It appears to us that according to Wertheim’s account (as per his 2008 paper), object motion should be underestimated in both cases (for self-motion in the opposite direction and in the same direction as the object). In both cases, one JND worth of “reference motion” would be subtracted from the object motion signal, leading to an underestimation of object motion. This is not what we found in our previous paper (Jörges & Harris, 2021), where we saw evidence for an overestimation of speed for observer and target motion in opposite directions, and no effect for observer and target motion in opposite directions. Omitting the work of Wertheim was, nonetheless, an oversight. We have added these references and thank the reviewer for pointing it out to us.

The arrows in figure 1 are unclear to me, why is noise increasing three times along the flow parsing pathway?

(1.6) It is meant to convey that, qualitatively, the self-motion-based estimate is noisier than the optic flow estimate, which then carries over to the estimate of object velocity and the predicted time-to-contact. We removed the additional (and confusing) noise icons.

I’m struggling a little bit with the word predicted. At least, it is not motion that is being predicted, it is time to contact that is being predicted. The motion is known, albeit presumably misperceived, and supposedly held constant to yield the time to contact estimate. This held constant assumption is critical for your analysis. This is sloppy around the end of page 3, for instance, but occurs throughout. Perhaps the term extrapolation fits better here?

(1.7) “Extrapolation” is indeed more precise, and we have changed “motion prediction” to “motion extrapolation” throughout, or changed the wording to “prediction of time-to-contact” or similar.

Relatedly, p4, line 4: remove the word motion from motion prediction, then the sentence makes sense to me.

P4 line 6, estimates, judgments may be a more appropriate word? Or percepts?

P4, line 25 and other links: please print the actual link in the article text. This way I is preserved better, e.g. in case the article is printed.

P4, line 35: “you”: rewrite

(1.8) Thank you, we addressed these points.

P5: line 6-7: rather critical sentence is incomplete, what happens after occlusion duration?

(1.9) Addressed. This sentence is only meant to express that the distance between the point of disappearance and the target depends on the speed of the object and the occlusion duration.

P5: why a gaussian velocity profile and not just a constant? This seems to me to complicate the situation. Assuming incomplete flow parsing, the object to be judged will also be seen to accelerate and decelerate (or vice versa) in the two movement conditions. What object speed is then used for the judgment, some kind of (weighted) average?

(1.10) The intention behind a Gaussian motion profile was to generate a more natural stimulus that could also be used on a MOOG motion platform in later experiments with physical motion. However, we agree with the reviewer in that it introduces some unnecessary complexity and have changed the self-motion profile to a constant speed of 4 m/s.

Do participants receive instructions about head motion? What happens to their view of the virtual world when they move their head, does it counterrotate so that the virtual environment is perceived as rigid? Would head movement make your data harder to analyze / add additional unwanted variability to your study (e.g., it could conceivably differ between conditions, although I am not directly aware of studies suggesting that the lateral simulated self-motion will induce head motion).

(1.11) The program is designed such that head motion occurs indeed relative to the scene, e.g., theoretically the person could turn all the way around and the visual scene would be behind them. We assume that any head movements would add only random variability; however, a literature search has not revealed any studies conducted to support this assumption. We have therefore added a safeguard to the experiment: we will instruct them to keep their heads still and to ensure they do this we will display a circle in the middle of the field of view that moves with their head. The circle disappears when it is close enough to the fixation cross (+-2.5°), and trials will only start when this condition is met i.e., when their head is aligned. We will also record the participant’s head rotation throughout the experiment. 

P5, line 23: is your task doable at very short durations (0.1 and 0.2 s especially)?

(1.12) While 0.1s and 0.2s are certainly with the range of reaction times for some participants, they can start planning their response before the ball disappears because the target rectangle is visible throughout the whole trial. This should allow them to complete the task successfully even at very short occlusion durations. 

I see that the ball casts a shadow on the ground. Is that a deliberate choice? Speed of the shadow over the tiles on the ground (relative speed between the two) is a direct cue to ball speed that could be used more straightforwardly than motion of the ball itself.

(1.13) We opted to simulate the shadow to embed the ball more firmly in the visual scene, but it is true that the relative motion between shadow and tiles provide a more easily interpretable cue to object motion. We have therefore decided to remove the shadow.

Speed estimation: describe more clearly that its only the balls in the ball cloud that move, not the whole scene.

(1.14) We added a sentence to make this explicit.

Is 37 trials sufficient for a JND estimate? Simulation work by Prins on his Bayesian staircase suggested you need more like 100 or so, if I recall correctly.

(1.15) Please note that we run two staircases per condition, which brings the number of trials per condition closer to Prins’ suggestion. Further, adding more trials should only lower the variability of the measured JNDs per condition and participant. Since we are interested in the effect of our manipulation on the population level, we can counteract this increased variability on the condition-per-participant level by testing more participants. Our power analysis shows that a combination of 37 trials per PEST (i.e., 74 per condition) and 40 participants is sufficient.

Are participants able to fixate the cross when it is so close to the rapidly moving ball cloud?

(1.16) From our personal, subjective experience with the stimulus, the eye is sometimes drawn to the big moving ball, while the dot cloud is less problematic. However, we think that this is unlikely to differ between the self-motion conditions and tasks, and therefore it should not do more than add random (albeit unwanted) variability. Allowing the observer to view the scene freely is likely to lead them to pursue the target with their gaze, which may lead to an Aubert-Fleischl-induced underestimation of the target speed. This should also be similar across conditions and tasks, but would, in our opinion, just introduce more unwanted variability.

Do you experience induced motion in the fixation cross during simulated self-motion? May the opposite direction of this induced motion confound your results?

(1.17) In a previous study (Jörges & Harris, 2021) with a setup similar to the one in the proposed study, we found no evidence that a moving textured background induced motion in the target. While we did not use equivalence testing or a Bayesian analysis to confirm the absence of an effect, the sample size was relatively large (30 participants and around 50 trials per condition); that is, we expect any effect of induced motion should be very small. In contrast to our previous study, we also opted to keep the background blank (see Fig 2), precisely to minimize the risk of induced motion affecting performance.

Do you use the same participants for the two tasks? Results such as Niehorster & Li 2017 suggest there may be wide variation between participants in how complete flow parsing is. That should give you ample variability between participants to do a strong test of correlation for you H3.

(1.18) Yes, the same participants will complete both tasks, and thank you again for drawing our attention to the Nierhoster & Li paper.

P 7, line 18-22. Why you assume that flow parsing is incomplete only in the opposite motion condition, but not in the same motion condition? That needs to be justified. Same for the precision prediction.

(1.19) This is based on (surprising) previous findings (Jörges & Harris 2021): in this study, we only found differences in perceived speed when observer and ball were moving in opposite directions, while the perceived speed for observer motion in the same direction as the ball were similar to the static condition. Similarly, we saw descriptive, but not statistical differences in precision between the opposite directions and the static conditions, while precision in the same direction was again very similar to precision in the static condition.

P 8, assumption atop the page: and you assume the distance is perceived correctly and is not affected by background motion. Can you justify this? If both perceived v and d may vary between conditions, you have a problem.

(1.20) The presence of motion parallax as a cue during self-motion might lead participants indeed to estimate depth differently, which in turn might affect the perceived physical distance between the point of disappearance and the target in the prediction task. Depth is usually underestimated in VR, so the added motion parallax might make performance more accurate, i.e., participants might perceive the stimulus as further away during self-motion than when they are static. When the same retinal stimulation is interpreted as stemming from an object that is further away, its physical speed would be underestimated independently of whether object and observer move in the same or in opposite directions. This is not what we found in our previous study (Jörges & Harris, 2021): in the condition most similar to this experiment, we found an overestimation of speed when observer and object moved in opposite directions, and no discernible difference between a static observer and observer and object motion in the same direction, where we would have predicted an underestimation of object speed. While motion parallax might have some effect, its impact seems to be negligible when compared to other sources of variability.

Furthermore, since the task is presented continuously in the same environment, with the stimuli at the same, constant distance, it is in our opinion likely that participants build up one consistent depth estimate across all trials and all parts of the trajectory, which might be used as a prior to interpret optic flow information.

Finally, the participants come to a full halt in the moment the object disappears; that is, in principle they are able to judge the distance between the point of disappearance and the target in the prediction task while stationary in every self-motion condition. 

Reviewer #2:

The prediction mentioned in the abstract makes perfect sense, but I feel that it is a very weak prediction: that there will be a correlation between performance in the two tasks. Specifying what one expects to be correlated might change this. I think it is a bit trivial that performance on the two tasks across different speeds of self-motion is correlated, but maybe the authors are referring to correlations across participants within each value of self-motion. Otherwise, maybe it makes more sense to check whether the values are similar, rather than only whether they are correlated (as in de la Malla et al., 2018, Errors in interception can be predicted from errors in perception. Cortex 98, 49-59).

(2.1) For Hypothesis 3, we are indeed looking for a correlation between performance in the speed estimation task and in the prediction task. However, we agree with the reviewer that merely finding a significant correlation between both tasks is a minimum requirement to support our hypothesis and perhaps not very informative on its own. To better assess not only if performance is correlated across both tasks, but also to which extent, we will use the models that generate our predictions to fit parameters that encode the effect of self-motion on perceived speed (separately for accuracy and precision) as a fraction of the self-motion speed. With that, they are scaled in the same way across both tasks, and any deviation from unity can be interpreted as either one task being affected more strongly by self-motion than the other. Reminiscent of de la Malla and her colleagues (2018), we can perform a regression between the fitted parameter in the prediction task and the fitted parameter in the speed estimation task and compute 95% confidence intervals, our prediction being that this confidence interval includes 1. To make this process more transparent to the reader, we have added a significant amount of detail information on the model and the fitting process in the manuscript.

I actually see a theoretical complication in interpreting the data. Since the self-motion presumably shifts the goal (the target rectangle) as much as it does the ball, why would you expect any bias in judging self-motion to influence the timing of the tap? I think this needs to be explained.

(2.2) Self-motion stops once the ball disappears, so in principle, participants should only be biased in estimating the speed of the object, not in estimating the distance between the point of disappearance and the target rectangle. One of the future avenues we see for this research is to introduce self-motion in different parts of the trial and compare to what extent the prediction of time-to-contact reflects biases in speed estimation (when self-motion occurs while the target is visible), in spatial updating (self-motion while the target is invisible), and both (self-motion throughout the whole trajectory). 

Another issue that needs justification is the use of a fixation point. Apart from making the task quite unnatural, it also introduces many complications. First of all, how will fixation be ensured. It is very difficult to keep fixating while making judgments about moving targets, and small periods of pursuit at critical moments might influence one’s judgments. Secondly, the participants might make several of the judgments with respect to the fixation point. The fixation point does not move with the simulated self-motion, so its motion relative to the surrounding also needs to be interpreted. It also provides a reference in time for the button press: the time it took the ball to reach/cross fixation. At the very least the authors should explain why they have a fixation point, and how this might influence their results. I would consider not requiring fixation.

(2.3) When participants follow the target with their gaze, they might underestimate the target’s speed due to the Aubert-Fleischl phenomenon (c.f., Dichgans, Wist, Diener, & Brandt, 1975). While this shouldn’t affect the results in a way that confounds the experiment as it should only lower the perceived speed across all conditions, requiring fixation curtails this problem altogether. Since we want to have the option to collect data remotely, we can’t collect our participant’s eye-movements to ensure continued fixation and have to trust that our participants fixate as instructed. The fixation cross is always located right in front of the observer and does move with them during self-motion. We have made this clearer in the manuscript, and we have also provided the rationale for including the fixation cross.

We understand and agree with the concerns raised by the reviewer, but we believe that preventing Aubert-Fleischl-like effects from occurring outweighs these concerns.

When the authors write “the process of flow parsing should add noise and lead to object speed judgements being more variable during self-motion” they are actually making some assumptions. Although these assumptions are probably reasonable, I think the authors should be explicit about the details. Assuming that people use some kind of flow parsing mechanism to separate object motion from self-motion, they presumably also have to do so when there is no self-motion. Thus, the assumption is that speed judgments become more noisy when self-motion is faster, just as they become more noisy (at least in absolute terms; it could be a fixed Weber fraction) when the object moves faster. Being very explicit about the assumptions will help the reader follow the reasoning. 

(2.4) Thank you and we made this more explicit in the introduction.

It might also be worthwhile more explicitly considering the consequences of the visual self-motion information being in conflict with information from other sources. Following Figure 1, perceived self-motion should be weak because 3 of the 4 ‘senses’ of the multisensory integration indicate that there is no self-motion.

(2.5) In our view, under most theories of multisensory integration this would depend on the weighting each cue receives (be it according to their relative reliabilities or other). We made a note of this in the introduction, but please note that we cut down on the section on multisensory perception of self-motion at the suggestion of reviewer #1, as we don’t have any commitment to a specific theory of multisensory integration of self-motion cues beyond self-motion speed being underestimated at least slightly when only visual cues are presented.

There is no evident reason for an asymmetry between motion with or against the ball.

(2.6) There is indeed no evident reason, but we have found evidence for such asymmetry that we consider rather compelling in a previous study (Jörges & Harris, 2021). While we will certainly verify statistically whether this asymmetry also holds for the prediction task (and is consistent between speed estimation and time-to-contact judgements), it seems reasonable to assume that the asymmetry will hold up for the purposes of planning the present study.

I am also not so sure about this interpretation of ‘flow parsing’. Flow parsing refers to the ability to separate object motion from self-motion from the visual information alone. That is indeed necessary for the proposed processing, but I don’t think that a multisensory value of self-motion is normally considered as an input to flow parsing, so maybe the terminology should be adjusted here.

(2.7) We have rephrased this section in the introduction and made clear that the original flow parsing hypothesis refers to a purely visual mechanism.

Actually, many of the claims and assumptions do not appear to be necessary for answering the question as to whether biases in speed estimation give the anticipated errors in prediction, so probably the introduction (and methods) can be simplified. 

(2.8) At the reviewer’s (and reviewer #1’s) suggestion, we have removed some of the detail on the multisensory perception of self-motion and integrated it with the paragraph on flow parsing.

Moreover, the last pair of hypotheses are what the authors really want to test (I think). They need to check that their manipulation (simulated self-motion) influences judged object speed (and its variability) but actually they already know that it will. Hypothesis 2 is therefore a bit superfluous. They plan to examine whether motion prediction is also influenced (Hypothesis 1) and whether it is influenced in the same manner (hypothesis 3). If it is influenced in the same manner, it must be influenced, so hypothesis 1 is also superfluous. This gives a much clearer study with one hypothesis (with two components: bias and variability).

(2.9) We treat Hypothesis 2 partially as a replication, and partially as a more highly powered attempt to detect an effect of self-motion on variability. Hypothesis 1 aims to establish whether self-motion affects time-to-contact estimation, which seems likely, but shouldn’t be assumed a priori. Hypothesis 3 builds on Hypotheses 1 and 2, but could be true even if Hypotheses 1 and/or 2 are not supported: e.g., self-motion could affect participants in similar ways across both tasks, without there being a consistent population-wide effect in either direction. In this case, we would see the high correlation we expect if Hypothesis 3 is true without Hypotheses 1 and 2 being supported. It is also possible that participants show the expected effect of self-motion in both tasks on average, but that this effect is not consistent for each participant across both tasks. It is therefore our view that each of our hypotheses, while they of course build on each other and interact with each other, has merit of its own and should be treated separately.

There are also a number of things to consider in the methods. Especially if people will be tested at home, the authors might want to consider the extent to which participants are allowed to move their heads, and whether such head movements will be compensated for.

(2.10) Please see our response 1.11 to reviewer #1.

Why are stereoblind participants excluded? Do the authors expect their performance to be different? 

(2.11) Stereovision is likely involved in judging the depth of the stimulus. While it seems most likely that stereo-blindness would only lead to unwanted, yet random variability rather than biases, administering stereotests can help reduce this variability at a very minor cost.

Is it a good idea to always center the trajectory in front of the observer, especially when that position is indicated by a fixation point? Maybe the authors should consider adding some jitter to the position. Otherwise the task could be performed by pressing the button after the same time from when the ball reaches fixation as the time between the ball appearing and it reaching fixation.

(2.12) Only the visible part of the trajectory is centered in front of the observer, and the target disappears by itself, without the observer’s intervention. We further use three different occlusion intervals (0.5, 0.6 and 0.7s), which should prevent the observer from circumventing the actual task by using heuristics.

It appears to me from the video that the target disappears when the participant presses the button. Is that correct? This should be mentioned explicitly. Since the task is to press the button when the ball would hit the target, this task could be interpreted as judging the time of collision of two moving items, rather than in terms of self-motion. If the target’s motion is underestimated due to motion in the surrounding one might therefore find no effect even though the hypothesis is true. Is there some reason to exclude this possibility? 

(2.13) The target disappears after 0.5s of motion independently of the observer’s actions. The observer’s only task in the prediction experiment is to press the button when the ball would have hit the target rectangle (had it not disappeared). Self-motion occurs only while the ball is visible, that is, self-motion should not bias the estimated distance the ball has to cover between the point of disappearance and the target rectangle.

Why was this velocity profile chosen for the self-motion? Not having a constant speed means that the response could be different for the two tasks simply because the moment that is considered relevant is different: for judging speed, presumably only the average speed is relevant, whereas for prediction the change in speed is presumably also relevant.

(2.14) Please see our response 1.10 to reviewer #1.

I assume that the training on the prediction task was always with the observer static. This should be specified. 

(2.15) The reviewer is correct in their assumption, and we have made this clearer in the manuscript.

Why is there no target in the speed estimation task (in the condition with a single ball)? Might this not influence the comparison? 

(2.16) We added the target to the speed estimation task. You can watch a video (also included in the instruction video) here: https://youtu.be/AsROlBXzgr0

Nice instruction video! I assume you also have a version with the other order.

(2.17) Thank you, and we do, yes. We have added links to both versions to the paper.

The status of the assumptions in the predictions section is not quite clear. Some of the assumptions are predictions based on earlier findings, but if the current results turn out to be slightly different it is not a problem. For instance, if the speed of the ball is overestimated by 30% rather than 20% at this speed (or the Weber fraction is not 10%) the reasoning will still hold in the same manner. In the case of the variability it might even be a problem if the results were identical to the previous ones (no influence of self-motion). The third assumption is very philosophical. How would you know whether they have the same bias other than by comparing performance in the two tasks, which is what the study was planned to examine so it cannot be an assumption. The same is true for Equation 3. The Weber fraction of 5% for distance judgments is presumably really an assumption that must be considered when converting speed judgment uncertainty into temporal uncertainty using Equation 3. Maybe explain exactly how this is done and therefore how sensitive the result is to deviations from this value.

(2.18) These are assumptions that we make in order to simulate datasets, both to visualize what we expect the data to be collected to look like, and to conduct the power analyses. As such, we have to make assumptions about the effect size we are expecting to observe, like the assumption that the same biases and variability differences are at play in both tasks, that the perceived speed is biased by 20% of the self-motion speed, etc. Please note that our experiment is specifically intended to test these assumptions about the effect of self-motion on perceived speed. We have added a more detailed description of the way we model the expected data in Appendix A.

To address the reviewer’s sample question directly: Since there are no variability-weighted components in this model, deviations from the expected Weber Fractions should only lead to non-systematic changes in precision across all conditions.

In the motion prediction section I think it would be a good idea to clarify that certain predictions are based on earlier research, while others are based on reasoning. This might be important for the interpretation, because not finding the asymmetrical influence of background motion, for instance, need not affect the general conclusion, whereas not finding an increase in the standard deviation with the magnitude would make some of the proposed analysis meaningless.

(2.19) We have made some of the assumptions more explicit and marked clearly which are based on the literature, which relate to our hypotheses, and for which we have found support ourselves in our previous study (Jörges & Harris, 2021).

In Figure 5A I am guessing the y-axis should be in s, not m/s. Why do the authors anticipate precisely this relationship? I think the authors can be a bit more specific about the actual values. Presumably these duration values are obtained by multiplying the difference in PSE by the occlusion time, or something like that. I would be specific, because that is what makes pre-registration a powerful tool. 

(2.20) Figure 5A shows the expected relationship between biases induced by self-motion in the speed estimation task (x axis) and the prediction task (y axis, which should indeed be in s rather than m/s – we thank the reviewer for spotting this), respectively. These predictions emanate from the models outlined under “Predictions”. For statistical testing, what matters is that we expect a negative relationship between the self-motion bias in the speed estimation task (operationalized as mean difference between the PSEs in the “Opposite Directions” condition and the “Static” condition) and the self-motion bias in the prediction task (operationalized as the mean difference between the timing error in the “Opposite Directions” self-motion condition and the ”Static” self-motion condition”).

As stated above, we agree that a mere (negative) correlation is a minimum finding, and we should indeed aim to quantify to what extent performance in one task predicts performance in the other. We have added a modelling section to the manuscript to address this issue.

Figure 5B also confused me. If the authors expect such a mess, why bother?

(2.21) The effect on precision (Figure 5B) looks indeed quite weak (visually). However, please note that the difference in variability underlying this plot is relatively big (an increase in variability of 20% of the presented speed). The issue here is that there are other sources of variability that, to some extent, obscure this effect in the measurement we take. We compensate for this high degree of noise by testing a much larger number of participants than we would test if we were only interested in the effect of self-motion on accuracy (see also the discrepancy between hypotheses relating to accuracy and hypotheses relating to precision in the power analysis).

I am not very familiar with the Wilkinson & Rogers notation, so I may be wrong, but it appears from Equation 4 that the authors assume linear, independent effects of observer motion, ball motion and occlusion duration. Why? Would you not for instance expect a larger effect of speed for a longer occlusion duration?

(2.22) By setting up the linear mixed models this way, we assume that each participant will show a linear effect of the ball speed on the timing error. We further assume no interactions between any of the effects. We agree that the assumption that observer motion, ball speed and occlusion direction interact is reasonable. However, the idea behind the random effects in a multilevel regression model (i.e., the effects specified in brackets) is to capture as much variability as possible in order to prevent uncaptured variability from biasing the regression coefficients of interest (in our case the regression coefficients pertaining to observer motion) and to raise statistical power. While it is generally desirable to match the expected structure of the variability as closely as possible, a large number of random effects (like a three-way interaction between observer motion, ball speed, and occlusion duration) can lead the model to become hard or even impossible to fit to the data. The structure we have chosen is a compromise between capturing the most important variability components and being able to fit the model. We have also verified its adequacy by performing a “power” analysis (or more accurately a “false positive analysis”) under the assumption of no effect, which yielded a false positive rate at the expected level of alpha = 0.05.

Just under that equation the authors speak of biases in timing error. Do they mean systematic errors? This is not really a bias but a potential finding: that observer motion influences timing errors. 

(2.23) Indeed, this is one of the statistical hypotheses we are testing. It seems to us that our use of the term “bias” is equivalent to what the reviewer understands as “systematic errors”: mean differences in response to a manipulation, or differences in accuracy. This is the conventional use of the term “bias” in studies of this kind.

What would not finding such an effect mean? Maybe the target position is shifted to the same extent as the ball, so their effects cancel?

(2.24) Not finding an effect of the self-motion profile in this analysis would, per se, provide evidence that participants can extrapolate motion with a reasonable degree of accuracy even when they experience visual self-motion while they observe the object. What this means then depends on our findings for the other hypotheses: if we replicate our earlier findings (Jörges & Harris, 2021) and find that the participants in this study also overestimate the speed of objects they observe while experiencing self-motion in the opposite direction, then a negative result for hypothesis (1a) would mean that participants use other sources of information in the prediction task than in the speed estimation task (e.g., prior information from “Static” trials) or perhaps that information is processed differently in a purely perceptual task than in an action-oriented task. This would be a surprising result that we would discuss thoroughly in the discussion, and which would certainly call for further investigation.

In our opinion, the specific alternate explanation the reviewer mentions, namely that the effects of self-motion on perceived ball speed and target position might cancel out, is not quite as likely as the observer comes to a halt in the moment in which the ball disappears, such that they should be able to estimate the distance between the point of disappearance and the target rectangle accurately.

I see many potentially interesting issues to explore, but the idea of pre-registration is to precisely specify what you are testing. For this, I think the authors need to better specify which effect they expect and why.

(2.25) We expect the regression coefficient for the self-motion profile “Opposite Directions” to be significantly different from zero and positive, which would constitute evidence that the timing error is more negative (i.e., participants pressed the button too early) in the “Opposite Directions” self-motion profile than when participants are static. We have made our expectations explicit for all statistical tests mentioned in the analysis section of the manuscript.

For equations 5 and 6 the measure is clear: all that matters is whether including the Motion Profile in the model provides a significant improvement. Finally, it seems that equation 9 is evaluating whether the judged speed influences the judged timing. Is this really what you want to know? Should you not be testing whether differences in judged speed can fully account for the differences in timing?

(2.26) We agree with that reviewer in that the statistical test expressed in equation 9 – which assesses whether differences in judged speed relate to timing errors at all – is a very low bar, and it would be more interesting to assess the degree to which perceived speed explains timing differences. We do this by modelling the results (see new sub-section on “Model fitting”).

By equations 10 and 11 you lost me completely. If the time difference and the JND difference are not independent this might give confusing results.

(2.27) We are indeed almost certain that mean differences in timing and JND differences will share a relevant amount of variability. However, our analysis establishes whether JND differences explain any variability in the precision of the timing responses beyond what is already explained by the mean differences in timing. We will only use the Likelihood Ratio Test as evidence here and we will not interpret the regression coefficients obtained in this analysis as they are, as the reviewer points out, are impossible to interpret. We have made this explicit in the analysis section.

In the power analysis I do not see any measure of the original assumed variability and effect size. Maybe I missed something.

(2.28) For complex nested designs such as ours (with several participants completing several repetitions of several conditions), analytic power analyses (such as the ones implemented in the program G Power, for example) are not possible. For this reason, we used the models outlined under “Predictions” to generate synthetic datasets based on what we have called “assumptions” in that section (e.g., the different within- and between-participant sources of variability that are likely to influence the dependent variable in the task, as well as the strength of the effect of self-motion on both accuracy and precision). We simulated 250 datasets for each combination of number of participants and number of repetitions per condition and performed all analyses (see Equations 5 to 11) over these synthetic datasets. The fraction of datasets for which these analyses return a significant result can then be taken as the power for each analysis.

Reviewer #3: 

On Page 2, line 21 you mention that “in many virtual reality (VR) applications, vestibular and proprioceptive cues signal that the body is at rest, while the visual optic flow cues simultaneously indicate self-motion.” Could you provide some examples and also explain why this is the case just in some VR applications but not others (e.g. is it due to properties of the hardware or the virtual environment itself?).

(3.1) At the suggestion of Reviewer #1, we reduced this paragraph and it no longer contains this reference to VR applications. We were referring to room-scale VR applications that allow the observer to walk around in the virtual environment on their own accord. But in fact any instance when you are watching a scene taken by a moving camera, e.g., on television or at the movies, provides this intersensory conflict.

On Page 3 lines 11-19 there seem to be references missing for some of the statements you make.

(3.2) While the first half of this paragraph is intended to draw the reader’s attention to some of the geometric properties of our setup, we added a reference (recommended by reviewer #1) that has explored some of these physical relationships (particularly those involving depth) and their impact on flow parsing.

On Page 3, Line 24 you say “it would seem logical that the prediction reflects this bias in motion estimation”. I would like to see a more detailed explanation for this assumption since it is critical for your study. I find that entire paragraph containing explanations that are a bit rushed and unclear.

(3.3) We have expanded on the processes at play (and possible scenarios in which predictions might not reflect these biases) in the introduction.

I am however not sure of the acceptance of any VR HMDs owned by participants. You mention that in-person testing would be conducted on a VIVE Pro Eye if granted permission, but it is expected that participants may possess different HMDs such as Quest 1 or 2, which have significantly different specifications and most importantly, interaction methods (e.g. controllers). Especially for time-sensitive stimuli that you present, it would be important to first conduct a pre-assessment of how your Unity code runs on these HMDs. I understand that due to COVID restrictions currently in place you would have to test remotely, but I believe more should be done to mitigate potential limitations arising from this. Perhaps one option would be to cap the framerate and field of view to certain parameters which are compatible with those HMDs that are lowest in terms of specifications that you would still accept in your study.

(3.4) We share the reviewer’s concerns and have limited the field of view in all programs to 80° (horizontally and vertically). Unity already caps the frame rate at 60 Hz, and drops below this framerate should occur almost never, as our programs require much less processing power than any VR applications the participants might use regularly. Our input method uses the keyboard for all participants to eliminate variability or technical issues due to different controllers.

For both tasks it is unclear how the speeds and sizes of the stimuli were determined. Was that based on previous literature? if so it should be mentioned or otherwise it should be based on piloting data.

(3.5) We have used a similar setup in our previous study (Jörges & Harris 2021), the only difference being that the single target and the ball in the ball cloud are now the same size. We made this change because we found that the ball cloud was judged consistently faster in the 2021 study, which we suspect to be a product of the difference in sizes between the single ball and the balls in the ball cloud in the previous study. The speeds and occlusion intervals were chosen such that, when participants keep their gaze on the fixation cross, the whole trajectory unfolds within a field of view of about 60° (of a total field of view of 80°), which is well within the field of view of any modern HMD. We have added these details in the methods section.

References:

Dichgans, J., Wist, E., Diener, H. C., & Brandt, T. (1975). The Aubert-Fleischl phenomenon: A temporal frequency effect on perceived velocity in afferent motion perception. Experimental Brain Research, 23(5), 529–533. https://doi.org/10.1007/BF00234920

Jörges, B., & Harris, L. R. (2021). Object speed perception during lateral visual self-motion. Attention, Perception, & Psychophysics.

---

## [Decision Letter · Decision Letter 1]

20 Apr 2022

The Impact of Visually Simulated Self-Motion on Predicting Object Motion – A Registered Report Protocol

PONE-D-21-37478R1

Dear Dr. Jörges,

We’re pleased to inform you that your manuscript has been judged scientifically suitable for publication and will be formally accepted for publication once it meets all outstanding technical requirements (note some of this text might not be well adapted for Registered Reports, so bear that in mind). All three reviewers replied with Accept, and I am also satisfied with the changes. Well done. Note two reviewers have included a few other suggestions you might find useful.

Kind regards,

Michael J Proulx, Ph.D.

Academic Editor

PLOS ONE

Additional Editor Comments (optional):

Reviewers' comments:

Reviewer's Responses to Questions

**Comments to the Author**

1. Does the manuscript provide a valid rationale for the proposed study, with clearly identified and justified research questions?

Reviewer #1: Yes

Reviewer #2: Yes

2. Is the protocol technically sound and planned in a manner that will lead to a meaningful outcome and allow testing the stated hypotheses?

Reviewer #1: Yes

Reviewer #2: Yes

3. Is the methodology feasible and described in sufficient detail to allow the work to be replicable?

Reviewer #1: Yes

Reviewer #2: Yes

4. Have the authors described where all data underlying the findings will be made available when the study is complete?

Reviewer #1: Yes

Reviewer #2: Yes

5. Is the manuscript presented in an intelligible fashion and written in standard English?

Reviewer #1: Yes

Reviewer #2: Yes

6. Review Comments to the Author

You may also provide optional suggestions and comments to authors that they might find helpful in planning their study.

Reviewer #1: The authors have done a very thorough and good job responding to my comments. I am looking forward to seeing the results of the study.

One further reference that may be of interest is https://jov.arvojournals.org/article.aspx?articleid=2770910

Reviewer #2: The purpose and methods are now much clearer. Most of my concerns clearly arose from having misunderstood details of the experiment. Many of the issues have been clarified sufficiently, but there are still a few small things that I believe will be helpful to clarify or motivate at some time.

The time course of the trials is now much clearer to me, which indeed solves many of my issues. I think it would be useful to modify figure 2C (and possibly D) to better match the actual experiment. This figure was the origin of much of my confusion, because in it the whole trajectory is more or less centred on fixation, rather than only the visible part. Probably the authors should also add something to illustrate when the observer moved (as now illustrated for the ‘invisible’ time). The pattern of events is quite complex, so it is good to have a reference. That might also help understand Table 1 which I still fail to understand, even when trying to work back from what it might mean. Is this the ball’s speed relative to fixation? I do not understand why the difference between ball speeds for the static observer changes in this manner. Or is there a typo somewhere? Should the first value be 20.4? Since the fixation point moves with the participant (but does not actually move) the ball moves in the opposite direction due to self-motion. That explains why opposite directions increases ball speed, but why this strange pattern for same speed? I feel that I am still missing something. It might help to always specify relative to what the motion is measured or described, because it is not intuitive. For instance, simulated self-motion does not correspond with retinal motion, because the participant is fixating a point that is moving along with the participant and is therefore static on the screen. The ball speed ‘should’ (according to the reasoning in the paper) be judged relative to the world, rather than relative to the observer. Are participants aware of this (is it part of the instruction)? In the timing task it is obvious because the self-motion affects the target, but in the speed judgment task this is not self-evident. All this could be problematic for the further interpretation, but not for the parts that are based on the authors’ previous work. I think being even clearer about the task and stimuli will make it easier for the reader to follow the reasoning.

Another issue that I had not always interpreted correctly is the role of the simulations and which parts of the methods are about simulations. I think it does make sense, but sometimes it is not clear to me whether the data presented in the figures are the outcome of the simulations, and sometimes it is not clear whether part of the analysis also applies to the simulations. I would try to clarify this. For instance, in the figure captions simply replace “Predicted data …” (Figure 3) by something like “Data from simulations based on …”. In the ‘data analysis plan’ indicate which parts (if any) also apply to the simulations. Maybe also change the order of some sections, because I think that the power analysis is based on the same (kind of) simulations as the predictions. Thinking logically I can guess what the authors did, but it is better to be told explicitly.

Details:

The order of the bars is incorrect in Figure 4: opposite directions in yellow (legend) but rightmost is same directions (caption). The power seems to be 0.75 for precision in speed estimation in Figure 6 (so less than 0.85).

I would explicitly mention that the fixation cross is static on the screen when mentioning that it moves with the observer (page 5 line 36). It is obvious when you think of it, but at this stage in the paper the reader does not need to think of this so it is worth pointing it out. In the next sentence I would also add the word ‘lateral’: The target is presented at a lateral distance that depends on the speed of the ball … The study is quite complex so it helps to guide the reader a bit.

The phrase “Given the on-going COVID-19 pandemic …” is probably no longer relevant.

7. PLOS authors have the option to publish the peer review history of their article (what does this mean?). If published, this will include your full peer review and any attached files.

Reviewer #1: No

Reviewer #2: No

---

## [Editor Report · Acceptance letter]

25 Apr 2022

PONE-D-21-37478R1 

The Impact of Visually Simulated Self-Motion on Predicting Object Motion – A Registered Report Protocol 

Dear Dr. Jörges:

I'm pleased to inform you that your manuscript has been deemed suitable for publication in PLOS ONE. Congratulations! Your manuscript is now with our production department. 

Kind regards, 

on behalf of

Dr. Michael J Proulx 

Academic Editor

PLOS ONE